# Average-Case Averages: Private Algorithms for Smooth Sensitivity and Mean Estimation

**Mark Bun**
Boston University
mbun@bu.edu

**Thomas Steinke**
IBM Research – Almaden
smooth@thomas-steinke.net

## Abstract

The simplest and most widely applied method for guaranteeing differential privacy is to add instance-independent noise to a statistic of interest that is scaled to its global sensitivity. However, global sensitivity is a worst-case notion that is often too conservative for realized dataset instances. We provide methods for scaling noise in an instance-dependent way and demonstrate that they provide greater accuracy under average-case distributional assumptions. Specifically, we consider the basic problem of privately estimating the mean of a real distribution from i.i.d. samples. The standard empirical mean estimator can have arbitrarily-high global sensitivity. We propose the trimmed mean estimator, which interpolates between the mean and the median, as a way of attaining much lower sensitivity on average while losing very little in terms of statistical accuracy. To privately estimate the trimmed mean, we revisit the smooth sensitivity framework of Nissim, Raskhodnikova, and Smith (STOC 2007), which provides a framework for using instance-dependent sensitivity. We propose three new additive noise distributions which provide concentrated differential privacy when scaled to smooth sensitivity. We provide theoretical and experimental evidence showing that our noise distributions compare favorably to others in the literature, in particular, when applied to the mean estimation problem.

## 1 Introduction

Consider a sensitive dataset $x \in \mathcal{X}^n$ consisting of the records of $n$ individuals and a real-valued function $f : \mathcal{X}^n \to \mathbb{R}$. Our goal is to estimate $f(x)$ while protecting the privacy of the individuals whose data is being used. *Differential privacy* [14] gives a formal standard of individual privacy for this problem (and many others), requiring that, for all pairs of datasets $x, y \in \mathcal{X}^n$ differing in one record (called *neighbouring* datasets and denoted $d(x, y) \leq 1$), the distribution of outputs should be similar for both inputs $x$ and $y$.

The most basic technique in differential privacy is to release an answer $f(x) + \nu$, where $\nu$ is instance-independent additive noise (e.g., Laplace or Gaussian) with standard deviation proportional to the *global sensitivity* $\mathsf{GS}_f$ of the function $f$. Here,

$$\mathsf{GS}_f = \max_{y,z \in \mathcal{X}^n \, : \, d(y,z) \leq 1} |f(y) - f(z)|$$

measures the maximum amount that $f$ can change across all pairs of datasets differing on one entry, including those which have nothing to do with $x$.

Calibrating noise to global sensitivity is optimal in the worst case, but may be overly pessimistic in the average case. This occurs in many statistical settings where $x$ consists of i.i.d. samples from a reasonably structured distribution and the goal is to estimate a summary statistic of that distribution. The main example we consider in this work is that of estimating the mean of a distribution given i.i.d. samples from it. The standard estimator for the distribution mean is the sample mean. However,

for distributions with unbounded support, e.g., Gaussians, the global sensitivity of the sample mean is infinite. Thus we consider a different estimator and a different measure of its sensitivity.

A more fine-grained notion of sensitivity is the *local sensitivity* of $f$ at the dataset $x$, which measures the variability of $f$ in the neighbourhood of $x$. That is,

$$\mathsf{LS}_f(x) = \max_{y \in \mathcal{X}^n \, : \, d(x,y) \leq 1} |f(y) - f(x)|.$$

However, naïvely calibrating noise to $\mathsf{LS}_f(x)$ is not sufficient to guarantee differential privacy. The reason is that the local sensitivity may itself be highly variable between neighbouring datasets, and hence the magnitude of the noise observed in a statistical release may itself leak information about $x$.

The work of Nissim, Raskhodnikova, and Smith [27] addressed this issue by identifying *smooth sensitivity*, an intermediate notion between local and global sensitivity, with respect to which one can calibrate additive noise while guaranteeing differential privacy. Smooth sensitivity is a pointwise upper bound on local sensitivity which is itself "smooth" in that its multiplicative variation on neighboring datasets is small. More precisely, for a smoothing parameter $t > 0$, the *t-smoothed sensitivity* of a function $f$ at a dataset $x$ is defined as

$$\mathsf{S}_f^t(x) = \max_{y \in \mathcal{X}^n} e^{-t \cdot d(x,y)} \cdot \mathsf{LS}_f(y),$$

where $d(x,y)$ denotes the number of entries in which $x$ and $y$ disagree. Noise distributions which simultaneously do not change much under additive shifts and multiplicative dilations at scale $t$ can be used with smooth sensitivity to give differential privacy.

In this work, we extend the smooth sensitivity framework by identifying three new distributions from which additive noise scaled to smooth sensitivity provides *concentrated differential privacy*. We apply these techniques to the problem of mean estimation, for which we propose the *trimmed mean* as an estimator that has both high accuracy and low smooth sensitivity.

## 1.1 Background.

Before describing our results in more detail, we recall the definition of differential privacy.

**Definition 1** (Differential Privacy (DP) [14, 12]). *A randomized algorithm $M : \mathcal{X}^n \to \mathcal{Y}$ is $(\varepsilon, \delta)$-differentially private ($(\varepsilon, \delta)$-DP) if, for all neighboring datasets $x, y \in \mathcal{X}^n$ and all (measurable) sets $S \subseteq \mathcal{Y}$,*

$$\mathbb{P}\left[M(x) \in S\right] \leq e^{\varepsilon} \mathbb{P}\left[M(y) \in S\right] + \delta.$$

We refer to $(\varepsilon, 0)$-differential privacy as pure differential privacy (or pointwise differential privacy) and $(\varepsilon, \delta)$-differential privacy with $\delta > 0$ as approximate differential privacy.

Given an estimator of interest $f : \mathcal{X}^n \to \mathbb{R}$ and a private dataset $x \in \mathcal{X}^n$, the randomized algorithm given by $M(x) = f(x) + \mathsf{GS}_f \cdot Z$ is $(\varepsilon, 0)$-differentially private for $Z$ sampled from a Laplace distribution scaled to have mean 0 and variance $2/\varepsilon^2$ (i.e., density $e^{-\varepsilon|z|} \cdot \varepsilon/2$). We use the smooth sensitivity in place of the global sensitivity. That is, we analyse randomized algorithms of the form

$$M(x) = f(x) + \mathsf{S}_f^t(x) \cdot Z$$

for $Z$ sampled from an admissible noise distribution.

The original work of Nissim, Raskhodnikova, and Smith [27] proposed three admissible noise distributions. The first such distribution, the Cauchy distribution with density $\propto \frac{1}{1+z^2}$ (and its generalizations of the form $\frac{1}{1+|z|^\gamma}$ for a constant $\gamma > 1$) can be used with smooth sensitivity to guarantee pure $(\varepsilon, 0)$-differential privacy. These distributions have polynomially-decaying tails and finitely many moments, which means they may be appropriately concentrated around zero to guarantee accuracy for a single statistic, but can easily result in inaccurate answers when used to evaluate many statistics on the same dataset. Unfortunately, inverse polynomial decay is in fact essential to obtain pure differential privacy. The exponentially-decaying Laplace and Gaussian distributions, which appear much more frequently in the differential privacy literature, were shown to yield approximate $(\varepsilon, \delta)$-differential privacy with $\delta > 0$.

A recent line of work [15, 5, 26, 4] has developed variants of differential privacy which permit tighter analyses of privacy loss over multiple releases of statistics as compared to both pure and approximate

differential privacy. In particular, the notion of concentrated differential privacy (CDP) [15, 5] has a simple and tight composition theorem for analyzing how privacy degrades over many releases while accommodating most of the key algorithms in the differential privacy literature, including addition of Gaussian noise calibrated to global sensitivity.

**Definition 2** (Concentrated Differential Privacy (CDP) [15, 5])**.** *A randomized algorithm* $M : \mathcal{X}^n \to \mathcal{Y}$ *is* $\frac{1}{2}\varepsilon^2$*-concentrated differentially private (*$\frac{1}{2}\varepsilon^2$*-CDP) if, for all neighboring datasets* $x, y \in \mathcal{X}^n$ *and all* $\alpha \in (1, \infty)$*,* $\mathrm{D}_\alpha\left(M(x)\|M(y)\right) \leq \frac{1}{2}\varepsilon^2\alpha$*, where* $\mathrm{D}_\alpha\left(P\|Q\right) = \frac{1}{\alpha-1}\log \mathbb{E}_{X \leftarrow P}\left[(P(X)/Q(X))^{\alpha-1}\right]$ *denotes the Rényi divergence of order* $\alpha$*.*

Another variant, Rényi Differential Privacy (RDP) [26], is closely related to CDP. Specifically, $\frac{1}{2}\varepsilon^2$-CDP is equivalent to the conjunction of an infinite family of RDP guarantees, namely $(\alpha, \frac{1}{2}\varepsilon^2 \cdot \alpha)$-RDP for every $\alpha \in (1, \infty)$. In particular, any CDP algorithm (such as those we present) is also an RDP algorithm.

It is natural to ask whether concentrated differential privacy admits distributions that can be scaled to smooth sensitivity while offering better privacy-accuracy tradeoffs than Cauchy, Laplace, or Gaussian.

## 1.2 Our Contributions: Smooth Sensitivity and CDP

As concentrated differential privacy is a relaxation of pure differential privacy, Cauchy noise and its generalizations automatically guarantee CDP. However, admissible distributions for CDP could have much lighter tails. (The full version [6] contains a lower bound showing that quasi-polynomial tails are necessary, whereas pure DP tails must be polynomial.) Nevertheless, it is not clear what distribution to conjecture would have these properties. In this work, we identify three such distributions with quasi-polynomial tails and show that they provide CDP when scaled to smooth sensitivity. Detailed statements and proofs appear in the full version [6].

**Laplace Log-Normal:**  The first such distribution we identify, and term the "Laplace log-Normal" $\mathsf{LLN}(\sigma)$, is the distribution of the random variable $Z = X \cdot e^{\sigma Y}$ where $X$ is a standard Laplace, $Y$ is a standard Gaussian, and $\sigma > 0$ is a shape parameter. This distribution has mean zero, variance $2e^{2\sigma^2}$, and satisfies the quasi-polynomial tail bound $\mathbb{P}\left[|Z| > z\right] \leq e^{-\log^2(z)/3\sigma^2}$ for large $z$. The following result shows that scaling Laplace log-Normal noise to smooth sensitivity gives CDP.

**Proposition 3.** *Let* $f : \mathcal{X}^n \to \mathbb{R}$ *and let* $Z \leftarrow \mathsf{LLN}(\sigma)$ *for some* $\sigma > 0$*. Then, for all* $s, t > 0$*, the algorithm* $M(x) = f(x) + \frac{1}{s} \cdot \mathsf{S}_f^t(x) \cdot Z$ *guarantees* $\frac{1}{2}\varepsilon^2$*-CDP for* $\varepsilon = t/\sigma + e^{3\sigma^2/2}s$*.*

Intuitively, additive scaling of $Z = X \cdot e^{\sigma Y}$ is handled by $X$ (i.e., $\mathrm{D}_\alpha\left(Z\|Z+s\right) \leq \alpha s^2 e^{3\sigma^2}/2$), while multiplicative dilations are handled by $Y$ after taking logarithms (i.e., $\mathrm{D}_\alpha\left(Z\|e^t Z\right) \leq \mathrm{D}_\alpha\left(\sigma Y\|t+\sigma Y\right) = \alpha t^2/2\sigma^2$). Group privacy handles additive-multiplicative combinations.

**Uniform Log-Normal:**  Second, $\mathsf{ULN}(\sigma)$, is the distribution of $Z = U \cdot e^{\sigma Y}$ where $U$ is uniformly distributed over $[-1, 1]$, $Y$ is a standard Gaussian, and $\sigma > 0$ is a shape parameter. It has mean zero and variance $\frac{1}{3}e^{2\sigma^2}$, and also has the tail bound $\mathbb{P}\left[|Z| > z\right] \leq e^{-\log^2(z)/2\sigma^2}$ for all $z \geq 1$.

**Proposition 4.** *Let* $f : \mathcal{X}^n \to \mathbb{R}$ *and let* $Z \leftarrow \mathsf{ULN}(\sigma)$ *with* $\sigma \geq \sqrt{2}$*. Then, for all* $s, t > 0$*, the algorithm* $M(x) = f(x) + \frac{1}{s} \cdot \mathsf{S}_f^t(x) \cdot Z$ *guarantees* $\frac{1}{2}\varepsilon^2$*-CDP for* $\varepsilon = t/\sigma + e^{3\sigma^2/2} \cdot \sqrt{2/\pi\sigma^2} \cdot s$*.*

**Arsinh-Normal:**  Our final new distribution is the "arsinh-normal" which is the distribution of $Z = \frac{1}{\sigma}\sinh(\sigma Y)$ where $Y$ is a standard Gaussian and $\sinh(y) = (e^y - e^{-y})/2$ denotes the hyperbolic sine function. This distribution has mean zero, variance $\frac{e^{2\sigma^2}-1}{2\sigma^2}$, and quasi-polynomial tails. We show that it gives CDP, albeit with a worse dependence on the smoothing parameter $t$.

**Proposition 5.** *Let* $f : \mathcal{X}^n \to \mathbb{R}$ *and let* $Z = \sinh(Y)$ *where* $Y$ *is a standard Gaussian. Then, for all* $s, t \in (0, 1)$*, the algorithm* $M(x) = f(x) + \frac{1}{s} \cdot \mathsf{S}_f^t(x) \cdot Z$ *guarantees* $\frac{1}{2}\varepsilon^2$*-CDP for* $\varepsilon = 2\sqrt{t} + 1.2s$*.*

We conjecture that the privacy analysis of the arsinh-Normal distribution can be improved to match (or better) the guarantees of our other two distributions.

## 1.3 Our Contributions: Private Mean Estimation

We study the following basic statistical estimation problem. Let $\mathcal{D}$ be a distribution over $\mathbb{R}$ with mean $\mu$ and let $x = (x_1, \ldots, x_n)$ consist of i.i.d. samples from $\mathcal{D}$. Our goal is to design a differentially private algorithm $M$ for estimating $\mu$ from the sample $x$.

The algorithmic framework we propose is as follows. We begin with a crude estimate on the range of the distribution mean, assuming $\mu \in [a, b]$.[1] Then we compute a trimmed and truncated mean of $x$:

$$f(x) = \left[ \frac{x_{(m+1)} + x_{(m+2)} + \cdots + x_{(n-m)}}{n - 2m} \right]_{[a,b]},$$

where $x_{(1)} \leq x_{(2)} \leq \cdots \leq x_{(n)}$ denotes the sample in sorted order (a.k.a. the order statistics) and $[y]_{[a,b]} = \min\{\max\{y, a\}, b\}$ denotes truncation to the interval $[a, b]$. In other words, $f(x)$ first discards the largest $m$ samples and the smallest $m$ samples from $x$, then computes the mean of the remaining $n - 2m$ samples, and finally projects this to the interval $[a, b]$. Then we release $M(x) = f(x) + \mathsf{S}_f^t(x) \cdot Z$ for $Z$ sampled from an admissible distribution for $t$-smoothed sensitivity. This framework requires picking a noise distribution $Z$, a trimming parameter $m \in \mathbb{Z}$ with $n > 2m \geq 0$, and a smoothing parameter $t > 0$.

Our mean estimation framework is versatile and may be applied to many distributions. We prove the following three illustrative results for this framework. Theorem 6 gives a strong accuracy guarantee under a correspondingly strong distributional assumption, whereas Theorem 8 gives a weaker accuracy guarantee under minimal distributional assumptions. Theorem 7 relaxes the light-tail assumption of Theorem 6 and shows the effect of this on the accuracy guarantee.

**Theorem 6** (Mean Estimation for Symmetric, Subgaussian Distributions). *Let $\varepsilon, \sigma > 0$, $a < b$, and $n \in \mathbb{Z}$ with $n \geq O(\log(n(b-a)/\sigma)/\varepsilon)$. There exists a $\frac{1}{2}\varepsilon^2$-CDP algorithm $M : \mathbb{R}^n \to \mathbb{R}$ such that the following holds. Let $\mathcal{D}$ be a $\sigma$-subgaussian distribution that is symmetric about its mean $\mu \in [a, b]$. Then*

$$\mathbb{E}_{X \leftarrow \mathcal{D}^n} \left[ (M(X) - \mu)^2 \right] \leq \frac{\sigma^2}{n} + \frac{\sigma^2}{n^2} \cdot O\left( \frac{\log((b-a)/\sigma)}{\varepsilon} + \frac{\log n}{\varepsilon^2} \right).$$

The first term $\frac{\sigma^2}{n}$ is exactly the non-private error required for this problem and the second term is the cost of privacy, which is a lower order term for large $n$.

**Theorem 7** (Mean Estimation for Symmetric, Moment-Bounded Distributions). *Let $\varepsilon > 0$ and $\underline{\sigma} > 0$ and $a < b$, and $n \in \mathbb{Z}$ with $n \geq O(\log(n(b-a)/\underline{\sigma})/\varepsilon)$. There exists a $\frac{1}{2}\varepsilon^2$-CDP algorithm $M : \mathbb{R}^n \to \mathbb{R}$ such that the following holds. Let $\mathcal{D}$ be a distribution that is symmetric about its mean $\mu \in [a, b]$ and has variance $\sigma^2$. Suppose $\sigma > \underline{\sigma}$ and $\mathbb{E}_{X \leftarrow \mathcal{D}} \left[ (X - \mu)^{2k} \right] \leq c_k^k \cdot \sigma^{2k}$ for some $k \in \mathbb{N}$ and $c_k \geq 1$. Then*

$$\mathbb{E}_{X \leftarrow \mathcal{D}^n} \left[ (M(X) - \mu)^2 \right] \leq \frac{\sigma^2}{n} + \frac{\sigma^2}{n^2} \cdot O\left( \frac{\log(n(b-a)/\underline{\sigma})}{\varepsilon} + \frac{c_k \cdot n^{1/k}}{\varepsilon^2} \right).$$

Note that a $\sigma$-subgaussian distribution satisfies the hypotheses of Theorem 7 with $c_k = O(k)$. Thus setting $k = \Theta(\log n)$ in Theorem 7 recovers the bound of Theorem 6.

**Theorem 8** (Mean Estimation for General Distributions). *Let $\varepsilon > 0$ and $\underline{\sigma} > 0$ and $a < b$. Let $n \geq O(\log(n(b-a)/\underline{\sigma})/\varepsilon)$. Then there exists a $\frac{1}{2}\varepsilon^2$-CDP algorithm $M : \mathbb{R}^n \to \mathbb{R}$ such that the following holds. Let $\mathcal{D}$ be a distribution with mean $\mu \in [a, b]$ and variance $\sigma^2 \geq \underline{\sigma}^2$. Then*

$$\mathbb{E}_{X \leftarrow \mathcal{D}^n} \left[ (M(X) - \mu)^2 \right] \leq \frac{\sigma^2}{n} \cdot O\left( \frac{\log(n(b-a)/\underline{\sigma})}{\varepsilon} + \frac{1}{\varepsilon^2} \right).$$

We stress that Theorems 6, 7, and 8 use the same algorithm; the only difference is in the setting of parameters and analysis. This illustrates the versatility of our algorithmic framework and that the distribution of the inputs directly translates to the accuracy of the estimate produced. We also

emphasize that, while the accuracy guarantees above depend on distributional assumptions on the dataset, the privacy guarantee requires no distributional assumptions, and holds without even assuming the data consists of i.i.d. draws.

In Section 4, we present an experimental evaluation of our approach when applied to Gaussian data.

## 1.4 Related Work

Prior works [4, 29] showed that, when scaled to smooth sensitivity, Gaussian noise provides the relaxed notion of *truncated* CDP [4] or Rényi DP [26], but does *not* suffice to give CDP itself. Other than this and the original distributions mentioned in Section 1.1, no other distributions have (to the best of our knowledge) been shown to provide differential privacy when scaled to smooth sensitivity.

We remark that smooth sensitivity is not the only way to exploit instance-dependent sensitivity. The most notable other example is the "propose-test-release" framework of Dwork and Lei [13]. Roughly, their approach is as follows. First an upper bound on the local sensitivity is proposed. The validity of this bound is then tested in a differentially private manner. If the test passes, then this bound can be used in place of the global sensitivity to release the desired quantity. If the test fails, the algorithm terminates without producing an estimate. This approach inherently requires relaxing to approximate differential privacy to account for the small probability that the test passes erroneously.

Dwork and Lei [13] apply their method to the trimmed mean to obtain asymptotic results (rather than finite sample bounds like ours). They obtain a bound on the local sensitivity of the trimmed mean using the interquantile range of the data, which is itself estimated by a propose-test-release algorithm. Then they add noise proportional to this bound. This requires relaxing to approximate differential privacy as well as assuming that the data distribution has sufficient density at the truncation points.

The mean and median (the extreme cases of the trimmed mean) have both been studied extensively in the differential privacy literature. We limit our discussion and study to the central model of differential privacy, though there has also been much work on mean estimation in the local model [21, 19, 11].

Nissim, Raskhodnikova, and Smith [27] analyze the smooth sensitivity of the median, but they do not apply it to mean estimation or give any average-case bounds for the smooth sensitivity of the median.

Smith [32] gives a general method for private point estimation, with asymptotic efficiency guarantees for general "asymptotically normal" estimators. This method is ultimately based on global sensitivity and, in large part due to its generality, does not provide good finite sample complexity guarantees.

Karwa and Vadhan [24] consider confidence interval estimates for Gaussians. Although they work in a different setting, their guarantees are similar to Theorem 6 (up to logarithmic factors). They propose a two-step algorithm: First a crude bound on the data is computed. Then the data is truncated using this bound and the mean is estimated via global sensitivity. We note that their algorithm does not readily extend to heavy-tailed distributions (like ours does, cf. Theorem 8). This is because a heavier-tailed distribution requires more conservative truncation to avoid distortion and hence their algorithm must add more noise due to the higher global sensitivity. Kamath, Li, Singhal, and Ullman [22] extend the work of Karwa and Vadhan to learning multivariate Gaussians.

Feldman and Steinke [16] use a median-of-means algorithm to privately estimate the mean, yielding a guarantee similar to Theorem 8.[2] Specifically, their algorithm partitions the dataset into evenly-sized subdatasets and computes the mean of each subdataset. Then a private approximate median is computed treating each subdataset mean as a single data point. This algorithm is simple and is applicable to any low-variance distribution, but (unlike our algorithm) its accuracy does not improve when the input data distribution is well-behaved.

The results of Karwa and Vadhan [24] and Feldman and Steinke [16] are most similar to ours. Our results (specifically, Theorems 6 and 8) roughly match theirs. However, our results apply to intermediate settings (e.g., Theorem 7 for, say, Student's T data), where the prior works do not achieve good accuracy. Our results can be viewed as providing a unified approach that *simultaneously* matches the results of Karwa and Vadhan [24] for Gaussians and Feldman and Steinke [16] for general distributions (and everything in between). The key advantage of our trimmed mean approach

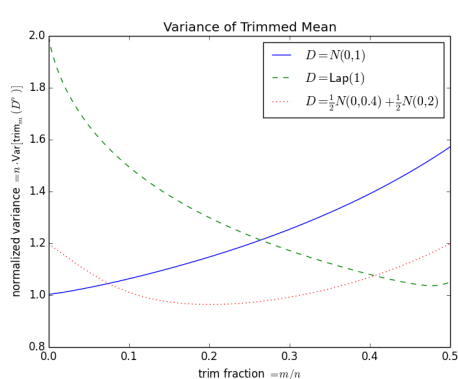

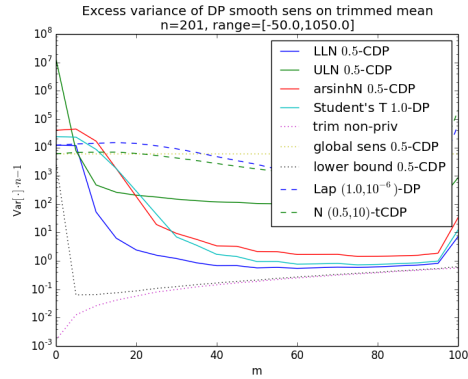

Figure 1: Variance of trimmed mean for various distributions as the trimming fraction is varied. The plot depicts $n = 1001$ averaged of $10^6$ runs.

Figure 2: Excesss variance of the private trimmed mean with smooth sensitivity. Data is $[N(0,1)^{201}]_{[-50,1050]}$. Average of $10^6$ runs.

is that the trimming automatically adjusts to the data distribution, whereas the prior approaches lack this versatility and rely on relatively brittle distributional assumptions.

Shortly after this work, Avella-Medina and Brunel [2] used the smooth sensitivity framework (and the propose-test-release framework) for *median* estimation. Of course, the mean and median are closely related. However, there is a subtle – but important – difference: Whereas the standard deviation provides the appropriate scale for the accuracy of an estimate of the mean, the reciprocal of the probability density around the median provides the appropriate scale for an estimate of the median [31]. Indeed, the standard deviation of the empirical mean and the empirical median scale with these quantities respectively. Accordingly, while our results state accuracy bounds in terms of the variance of the unknown distribution, their results state accuracy bounds in terms of the probability density in the neighbourhood of the median. Neither type of bound dominates the other, as it is easy to find distributions more favourable to each analysis. However, while their analysis and bounds are very different from ours, their algorithm is not; their algorithm is a special case of our algorithm. Thus we view their work as providing further independent validation of the utility of our approach.

**Further Applications**   Mean estimation is an extremely fundamental task that arises as a subroutine of more complex tasks. For example, private optimization and machine learning often rely on estimating gradients [3, 1]. This is a (multivariate) mean estimation task and our methods may yield improvements here. Mean estimation also naturally arises in hypothesis testing [33, 18, 7, 9, 10, 8].

The smooth sensitivity framework has also been applied to other problems. Examples include learning decision forests [17], principal component analysis [20], analysis of outliers [28], and analysis of graphical data [23, 25, 34, 30]. Our new distributions can immediately be applied to these problems.

After estimating the mean (or location parameter) of a distribution, the next question is to estimate its scale (e.g. variance). For this, our methods can be applied to robust location estimators [31].

## 2   Trimmed Mean

For the problem of mean estimation, we use the trimmed mean as our estimator.

**Definition 9** (Trimmed Mean). *For $n, m \in \mathbb{Z}$ with $n > 2m \geq 0$, define* $\mathsf{trim}_m : \mathbb{R}^n \to \mathbb{R}$ *by*

$$\mathsf{trim}_m(x) = \frac{x_{(m+1)} + x_{(m+2)} + \cdots + x_{(n-m)}}{n - 2m},$$

*where $x_{(1)} \leq x_{(2)} \leq \cdots \leq x_{(n)}$ denote the order statistics of $x$.*

Intuitively, the trimmed mean interpolates between the mean ($m = 0$) and the median ($m = \frac{n-1}{2}$).

**Error of the Trimmed Mean:**   Before we consider privatising the trimmed mean, we look at the error introduced by the trimming itself. We focus on mean squared error relative to the mean. That is, $\underset{X \leftarrow \mathcal{D}^n}{\mathbb{E}} \left[ (\mathsf{trim}_m(X) - \mu)^2 \right]$, where $\mu = \underset{X \leftarrow \mathcal{D}}{\mathbb{E}} [X]$ is the mean of the distribution $\mathcal{D}$.

We remark that mean squared error may not be the most relevant error metric for many applications. For example, the length of a confidence interval may be more relevant [24]. Similarly, the mean may not be the most relevant parameter to estimate. We pick this error metric as it is simple, widely-applicable, and does not require picking additional parameters (such as the confidence level).

The error of the trimmed mean depends on both the trimming fraction and also the data distribution. Figure 1 illustrates this. For Gaussian data, the optimal estimate is the empirical mean, corresponding to trimming $m = 0$ elements. This has mean squared error $\frac{1}{n}$ for $n$ samples. As the trimming fraction is increased, the error does too. At the extreme, the median of Gaussian data has asymptotic variance $\frac{\pi}{2n} \approx \frac{1.57}{n}$. However, if the data has slightly heavier tails than Gaussian data, such as Laplacian data, then trimming actually reduces error! The Laplacian mean has variance $\frac{2}{n}$, while the median has asymptotic variance $\frac{1}{n}$. In between these two cases is a mixture of two Gaussians with the same mean and differing variances. Here a small amount of trimming reduces the error, but a large amount of trimming increases it again, and there is an optimal trimming fraction in between.

For our main theorems we use the following analytic bound.

**Proposition 10.** *Let $n, m \in \mathbb{Z}$ satisfy $n > 2m \geq 0$. Let $X_1, \cdots, X_n$ be i.i.d. samples from a distribution $\mathcal{D}$ on $\mathbb{R}$ with mean $\mu$ and variance $\sigma^2$. Then*

$$\mathbb{E}\left[(\mathsf{trim}_m(X) - \mu)^2\right] \leq \frac{n(1 + \sqrt{8m})}{(n - 2m)^2}\sigma^2 = O\left(\frac{m}{n}\right)\sigma^2.$$

*Furthermore, if $\mathcal{D}$ is a symmetric distribution, then $\mathsf{trim}_m(X)$ is also symmetric and*

$$\mathbb{E}\left[(\mathsf{trim}_m(X))^2\right] \leq \frac{n}{(n - 2m)^2} \cdot \sigma^2 = \left(1 + O\left(\frac{m}{n}\right)\right)\frac{\sigma^2}{n}.$$

**Sensitivity of Trimmed Mean:** The trimmed mean also has low smooth sensitivity.

**Proposition 11.** *Let $a, b, t \in \mathbb{R}$ with $a < b$ and $t \geq 0$ and $n, m, k \in \mathbb{Z}$ with $n > 2m \geq 0$ and $x \in \mathbb{R}^n$. Denote $x_1, x_2, \cdots, x_n$ in sorted order as $x_{(1)} \leq x_{(2)} \leq \cdots \leq x_{(n)}$. The $t$-smooth sensitivity of the trimmed mean restricted to inputs in $[a, b]$ – that is, $\mathsf{trim}_m : [a, b]^n \to [a, b]$ – is*

$$\mathsf{S}^t_{\mathsf{trim}_m}(x) = \frac{1}{n - 2m} \max_{k=0}^{n} e^{-kt} \max_{\ell=0}^{k+1} x_{(n-m+1+k-\ell)} - x_{(m+1-\ell)},$$

*where we define $x_{(i)} = a$ for $i \leq 0$ and $x_{(i)} = b$ for $i > n$.*

This is a direct extension of the analysis of the smooth sensitivity of the median by Nissim, Raskhodnikova, and Smith [27]. There is a $O(n \log n)$-time algorithm for computing the smooth sensitivity.

## 3  Average-Case Mean Estimation via Smooth Sensitivity of Trimmed Mean

Having compiled the relevant tools, we turn to the problem of mean estimation in the average-case distributional setting. We have an unknown distribution $\mathcal{D}$ on $\mathbb{R}$ and our goal is to estimate the mean $\mu$, given $n$ independent samples $X_1, \cdots, X_n$ from $\mathcal{D}$. Our non-private comparison point is the (un-trimmed) empirical mean $\overline{X} = \frac{1}{n}\sum_{i=1}^{n} X_i$. This is unbiased – that is, $\mathbb{E}_{X \leftarrow \mathcal{D}^n}\left[\overline{X}\right] = \mu$ – and has variance $\mathbb{E}_{X \leftarrow \mathcal{D}^n}\left[(\overline{X} - \mu)^2\right] = \frac{\sigma^2}{n}$, where $\sigma^2 = \mathbb{E}_{X \leftarrow \mathcal{D}}\left[(X - \mu)^2\right]$. We make the (necessary) assumption that some loose bound $\mu \in [a, b]$ is known. Our results will only pay logarithmically in $b - a$, so this bound need not be tight.

In our situation the inputs may be unbounded. This means the trimmed mean has infinite global sensitivity and infinite smooth sensitivity. Thus we apply truncation to control the sensitivity.

**Definition 12** (Truncation). *For $a, b, x \in \mathbb{R}$ with $a < b$, define $[x]_{[a,b]} = \max\{\min\{x, b\}, a\}$. For $x \in \mathbb{R}^n$ and $a < b$, define $[x]_{[a,b]} = ([x_1]_{[a,b]}, [x_2]_{[a,b]}, \cdots, [x_n]_{[a,b]})$.*

**Truncation of Inputs:** By truncating inputs before applying the trimmed mean, we obtain the following error bound. This holds for symmetric and subgaussian distributions.

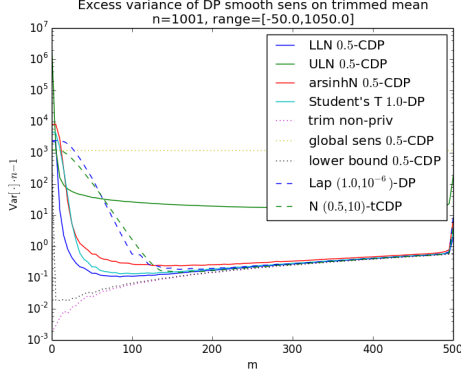

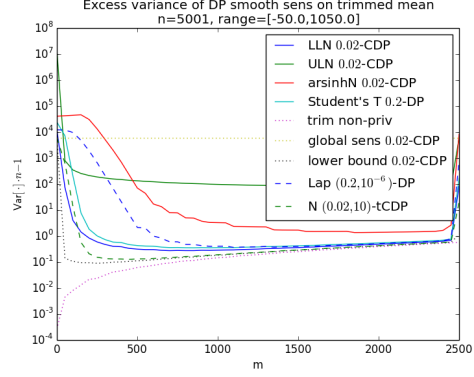

Figure 3: Excesss variance of the private trimmed mean with smooth sensitivity. Data is $[N(0,1)^{1001}]_{[-50,1050]}$. Average of $10^6$ runs.

Figure 4: Excesss variance of the private trimmed mean with smooth sensitivity. Data is $[N(0,1)^{5001}]_{[-50,1050]}$. Average of $10^6$ runs.

**Proposition 13.** *Let $\mathcal{D}$ be a symmetric $O(\sigma)$-subgaussian distribution on $\mathbb{R}$ with mean $\mu$ and variance $\sigma^2$. Let $a + O(\sigma \log n) < \mu < b - O(\sigma \log n)$. Let $n, m \in \mathbb{Z}$ satisfy $n > 3m \geq 0$. Then*

$$\underset{X \leftarrow \mathcal{D}^n}{\mathbb{E}} \left[ \left( \mathsf{trim}_m \left( [X]_{[a,b]} \right) - \mu \right)^2 \right] = \frac{\sigma^2}{n} \left( 1 + O\left( \frac{m}{n} \right) \right).$$

We remark that if $\mathcal{D}$ is not subgaussian, but rather subexponential, then a similar bound can be proved. Next we turn to analyzing the smooth sensitivity of the trimmed mean with truncated inputs.

**Lemma 14.** *Let $\mathcal{D}$ be a $\sigma$-subgaussian distribution on $\mathbb{R}$. Let $a < 0 < b$. Then*

$$\underset{X \leftarrow \mathcal{D}^n}{\mathbb{E}} \left[ \left( \mathsf{S}^t_{\mathsf{trim}_m\left( [\cdot]_{[a,b]} \right)}(X) \right)^2 \right] \leq \frac{8\sigma^2 \log n + e^{-2mt}(b-a)^2}{(n-2m)^2}.$$

*Proof.* By Proposition 11,

$$\mathsf{S}^t_{\mathsf{trim}_m\left( [\cdot]_{[a,b]} \right)}(x) = \frac{1}{n-2m} \max_{k=0}^{n} e^{-kt} \max_{\ell=0}^{k+1} x_{(n-m+1+k-\ell)} - x_{(m+1-\ell)}$$

$$\leq \frac{\max\{x_{(n)} - x_{(1)}, e^{-mt} \cdot (b-a)\}}{n-2m},$$

where the inequality follows from the fact that $x_{(n-m+1+k-\ell)} - x_{(m+1-\ell)} \leq x_{(n)} - x_{(1)}$ when $k < m$ and $x_{(n-m+1+k-\ell)} - x_{(m+1-\ell)} \leq b - a$ when $k \geq m$. Thus

$$\underset{X \leftarrow \mathcal{D}^n}{\mathbb{E}} \left[ \left( \mathsf{S}^t_{\mathsf{trim}_m\left( [\cdot]_{[a,b]} \right)}(X) \right)^2 \right] \leq \frac{1}{(n-2m)^2} \underset{X \leftarrow \mathcal{D}^n}{\mathbb{E}} \left[ (X_{(n)} - X_{(1)})^2 + e^{-2mt}(b-a)^2 \right]$$

$$\leq \frac{8\sigma^2 \log(2n) + e^{-2mt}(b-a)^2}{(n-2m)^2},$$

where the final inequality follows from properties of subgaussians [16, Lem. 4.5] and the fact that $(x-y)^2 \leq 4 \max\{x^2, y^2\}$ for all $x, y \in \mathbb{R}$. $\qquad \square$

Combining Proposition 13 and Lemma 14 with the distributions from Section 1.2 yields Theorem 6.

**Truncation of Outputs:** Rather than truncating the inputs to the trimmed mean, we can truncate the output. This is useful for heavier-tailed distributions and is also simpler to analyze: If Y is a random variable and $\mu \in [a,b]$, then $\mathbb{E}\left[ ([Y]_{[a,b]} - \mu)^2 \right] \leq \mathbb{E}\left[ (Y-\mu)^2 \right]$. Truncation of outputs also controls smooth sensitivity. An analysis analogous to that above yields Theorem 8.

# 4 Experimental Results

Figures 2, 3, & 4 show an experimental evaluation of our methods – specifically the combination of the trimmed mean with various smooth sensitivity distributions applied to Gaussian data. We briefly explain the experimental setup below. Please see the full version [6] for detail.

**Data & Error:** Our data is sampled from a standard univariate Gaussian distribution. The truncation interval is set conservatively to $[a, b] = [-50, 1050]$ and the data is truncated before applying the trimmed mean. We measure the variance or mean squared error of the various algorithms. I.e., $\sigma^2 = \mathbb{E}_{X \leftarrow N(\mu, 1)^n} \left[ \left( \text{trim}_m \left( [X]_{[a,b]} \right) + \mathsf{S}^t_{\text{trim}_m \left( [\cdot]_{[a,b]} \right)} (X) \cdot Z - \mu \right)^2 \right]$, where $Z$ is appropriate noise. For scaling, we multiply $\sigma^2$ by $n$ and subtract 1.

**Algorithms:** We compare our three noise distributions against three other distributions from prior work. Three further comparison points are provided. We explain each line briefly; see the full version [6] for more detail. `LLN`: The Laplace Log-Normal distribution. `ULN`: The Uniform Log-Normal distribution. `arshinhN`: The Arsinh-Normal distribution. `Student's T`: The Student's T distribution with 3 degrees of freedom. This is a simplification of a prior algorithm [27]. (The Cauchy distribution has infinite variance and is thus is not included.) `trim non-priv`: We plot the line where no noise is added for privacy. The only source of error is the trimmed mean itself. `global sens`: We plot the error that would be attained by truncating the data to $[a, b]$ and then using this to bound global sensitivity and add Gaussian noise. `lower bound`: We plot the lower bound on variance. `Lap`: Laplace noise. This was suggested in the original work [27]. `N`: Gaussian noise. This was analyzed in prior work [4] and provides the relaxed notion of truncated CDP.

**Privacy & Parameters:** The algorithms we compare satisfy different variants of differential privacy. As such, it is not possible to give a completely fair comparison. Our new distributions satisfy concentrated differential privacy, whereas the Student's T distribution satisfies pure differential privacy. Laplace and Gaussian noise satisfy approximate differential privacy or truncated concentrated differential privacy. To provide the fairest possible comparison, we pick a $\varepsilon$ value (namely, $\varepsilon = 1$ or $\varepsilon = 0.2$) and then compare $(\varepsilon, 0)$-differential privacy with relaxations thereof. Namely, we compare $(\varepsilon, 0)$-differential privacy with $\frac{1}{2}\varepsilon^2$-CDP, $(\frac{1}{2}\varepsilon^2, 10)$-tCDP, and $(\varepsilon, 10^{-6})$-differential privacy. Each of these is implied by $(\varepsilon, 0)$-differential privacy and the implication is fairly tight, so intuitively provides a roughly similar level of privacy. Aside from the privacy parameters ($\varepsilon$ etc.) and the dataset size ($n$), we show a range of trimming levels $m$ on the horizontal axis. We numerically optimize the smoothing parameter $t$. We set the distribution shape parameters to appropriate near-optimal values.

**Overall Performance:** The experimental results demonstrate that for relatively moderate parameter settings ($n = 201$ and $\varepsilon = 1$ depicted in Figure 2) it is possible to privately estimate the mean with variance that is only a factor of two higher than non-privately. For $n = 1001$, it is possible to drive this excess variance down to $10\%$. Indeed, in these settings, the additional error introduced by trimming is more significant than that introduced by the privacy-preserving noise!

We remark that the data for these experiments is perfectly Gaussian. If the data deviates from this ideal, the robustness of the trimmed mean may actually be beneficial for accuracy (and not just privacy). Figure 1 shows that for some natural distributions the trimming does reduce variance.

**Comparison of Algorithms:** The results show that different algorithms perform better in different parameter regimes. However, generally, the Laplace Log-Normal distribution has the lowest variance, closely followed by the Student's T distribution. The Arsinh-Normal distribution performs adequately, but the Uniform Log-Normal distribution performs poorly. The Laplace and Gaussian distributions from prior work perform substantially worse than our distributions in some parameter settings.

Note that the different algorithms satisfy slightly different privacy guarantees and also have very different tail behaviours. Since the variance of many of the algorithms is broadly similar, the choice of which algorithm is truly best will depend on these factors. If the stronger pure differential privacy guarantee is preferable, the Student's T distribution is likely best. However, this has no third moment and consequently heavy tails. This makes it bad if, for example, the goal is a confidence interval, rather than a point estimate of the mean. The lightest tails are provided by the Gaussian, but this only satisfies the weaker truncated CDP or approximate differential privacy definitions. Laplace Log-Normal is in between – it satisfies the strong concentrated differential privacy definition and has quasipolynomial tails and all its moments are finite.

## Acknowledgement

Part of this work was completed while the authors were at the Simons Institute for the Theory of Computing at the University of California, Berkeley.

## Footnotes

[1] Assuming an *a priori* bound on the the mean is necessary to guarantee CDP. However, such a bound can, under reasonable assumptions, be discovered by an $(\varepsilon, \delta)$-DP algorithm [24].

[2]The results of Feldman and Steinke [16] are stated for the (formally incomparable) problem of ensuring generalization for adaptive data analysis, but differentially private algorithms follow implicitly from their work.

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
