[Supplementary Material]

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

 here to work in the central model of differential privacy, which is the model we study, though there has also been much work on mean estimation in the local model of privacy [JKMW18; GRS18; DR19].

Nissim, Raskhodnikova, and Smith [NRS07] analyze the smooth sensitivity of the median, but they do not apply it to mean estimation or give any average-case bounds for the smooth sensitivity of the median.

Smith [Smi08] gave a general method for private point estimation, with asymptotic efficiency guarantees for general "asymptotically normal" estimators. This method is ultimately based on global sensitivity and, in large part due to its generality, does not provide good finite sample complexity guarantees.

Karwa and Vadhan [KV18] consider confidence interval estimates for Gaussians. Although they work in a different setting, their guarantees are similar to Theorem 6 (although weaker in logarithmic factors). They propose a two-step algorithm: First a crude bound on the data is computed. Then the data is truncated using this bound and the mean is estimated via global sensitivity. Kamath, Li, Singhal, and Ullman [KLSU19] provide algorithms for learning multivariate Gaussians (extending the work of Karwa and Vadhan). We note that their algorithm does not readily extend to heavy-tailed distributions like ours does (cf. Theorem 7).

Feldman and Steinke [FS18] use a median-of-means algorithm to privately estimate the mean, yielding a guarantee similar to Theorem 7. Specifically, their algorithm partitions the dataset into evenly-sized subdatasets and computes the mean of each subdataset. Then a private approximate median is computed treating each subdataset mean as a single data point. This algorithm is simple and is applicable to any low-variance distribution, but is not as accurate as our algorithm when the input data distribution is well-behaved. Our results (Theorems 6 and 7) can be viewed as providing a unified approach that *simultaneously* matches the results of Karwa and Vadhan [KV18] for Gaussians and Feldman and Steinke [FS18] for general distributions (and also everything in between).

The smooth sensitivity framework has been applied to other problems. Some examples include learning decision forests [FI17], principal component analysis [GGB17], analysis of outliers [OFS15], and analysis of graphical data [KRSY11; KNRS13; WW13].

# 2   Preliminaries

## 2.1   Differential Privacy

We refer the reader to Section 1.1 for the definitions of differential privacy and concentrated differential privacy.

The canonical CDP algorithm is Gaussian noise addition. Adding $N(0, \sigma^2)$ to a sensitivity-$\Delta$ function attains $\frac{\Delta^2}{2\sigma^2}$-CDP.

**Lemma 8.** *Let $\mu, \mu', \sigma \in \mathbb{R}$ with $\sigma > 0$. Then*

$$\forall \alpha \in (1, \infty) \quad \mathrm{D}_\alpha \left( N(\mu, \sigma^2) \big\| N(\mu', \sigma^2) \right) = \alpha \frac{(\mu - \mu')^2}{2\sigma^2}.$$

The relationship between concentrated differential privacy and pure/approximate differential privacy is summarized by the following lemma.

**Lemma 9** ([BS16]). *Let $M : \mathcal{X}^n \to \mathcal{Y}$ be a randomized algorithm. If $M$ satisfies $(\varepsilon, 0)$-differential privacy, then it satisfies $\frac{1}{2}\varepsilon^2$-CDP. If $M$ satisfies $\frac{1}{2}\varepsilon^2$-CDP, then it satisfies $\left( \frac{1}{2}\varepsilon^2 + \varepsilon \cdot \sqrt{2\log(1/\delta)}, \delta \right)$-differential privacy for all $\delta > 0$.*

We will make specific use of the following more precise statement of the conversion from pure to concentrated differential privacy.

**Proposition 10** ([BS16, Prop. 3.3]). *Let $P$ and $Q$ be probability distributions satisfying $D_\infty(P\|Q) \le \varepsilon$ and $D_\infty(Q\|P) \le \varepsilon$. Then $D_\alpha(P\|Q) \le \frac{1}{2}\varepsilon^2\alpha$ for all $\alpha \in (1,\infty)$.*

We also make use of the following group privacy property of concentrated differential privacy.

**Lemma 11.** *Let $P, Q, R$ be probability distributions. Suppose $D_\alpha(P\|R) \le a{\cdot}\alpha$ and $D_\alpha(R\|Q) \le b \cdot \alpha$ for all $\alpha \in (1,\infty)$. Then, for all $\alpha \in (1,\infty)$,*

$$D_\alpha(P\|Q) \le \alpha \cdot (\sqrt{a} + \sqrt{b})^2 \le 2\alpha \cdot (a+b).$$

*Proof.* We use the following triangle-like inequality for Rényi divergence.

**Lemma 12** ([BDRS18]). *Let $P, Q, R$ be probability distributions and $\beta, \gamma \in (1,\infty)$. Set $\alpha = \beta\gamma/(\beta + \gamma - 1)$. Then*

$$D_\alpha(P\|Q) \le \frac{\gamma}{\gamma - 1}D_\beta(P\|R) + D_\gamma(R\|Q).$$

We fix $\alpha$ and we must pick $\beta, \gamma \in (1,\infty)$ satisfying $\alpha = \beta\gamma/(\beta + \gamma - 1)$ to minimize

$$
\begin{aligned}
D_\alpha(P\|Q) &\le \frac{\gamma}{\gamma - 1}D_\beta(P\|R) + D_\gamma(R\|Q) \\
&\le \frac{\gamma}{\gamma - 1}a\beta + b\gamma \\
&= \frac{\alpha\gamma}{\gamma - \alpha}a + b\gamma \\
&= \alpha \cdot \left(\frac{u}{u - 1}a + bu\right),
\end{aligned}
$$

where the final equalities use the fact that $\beta = \alpha \cdot \frac{\gamma - 1}{\gamma - \alpha}$ and the substitution $\gamma = u\alpha$. We set $u = 1 + \sqrt{a/b}$ to minimize:

$$
\begin{aligned}
D_\alpha(P\|Q) &\le \alpha \cdot \left(\frac{u}{u - 1}a + bu\right) \\
&= \alpha \cdot \left(\frac{1 + \sqrt{a/b}}{\sqrt{a/b}}a + b(1 + \sqrt{a/b})\right) \\
&= \alpha \cdot \left(a + b + 2\sqrt{ab}\right) \\
&= \alpha \cdot \left(\sqrt{a} + \sqrt{b}\right)^2 \\
&\le \alpha \cdot \left(\sqrt{a} + \sqrt{b}\right)^2 + \alpha \cdot \left(\sqrt{a} - \sqrt{b}\right)^2 \\
&= \alpha \cdot (2a + 2b).
\end{aligned}
$$

$\square$

## 2.2 Smooth Sensitivity

The smooth sensitivity framework was introduced by Nissim, Raskhodnikova, and Smith [NRS07]. We begin by recalling the definition of local (and global) sensitivity, as well as the local sensitivity at distance $k$.

**Definition 13** (Global/Local Sensitivity). *Let $f : \mathcal{X}^n \to \mathbb{R}$ and $x \in \mathcal{X}^n$. The local sensitivity of $f$ at $x$ is defined as*

$$\mathsf{LS}_f(x) = \sup_{x' \in \mathcal{X}^n : d(x,x') \leq 1} |f(x') - f(x)|,$$

*where $d(x',x) = |\{i \in [n] : x'_i \neq x_i\}|$ is the number of entries on which $x$ and $x'$ differ. For $k \in \mathbb{N}$, the local sensitivity at distance $k$ of $f$ at $x$ is defined as*

$$\mathsf{LS}_f^k(x) = \sup_{x' \in \mathcal{X}^n : d(x',x) \leq k} \mathsf{LS}_f(x') = \sup_{x',x'' \in \mathcal{X}^n : d(x,x') \leq k, d(x',x'') \leq 1} |f(x'') - f(x')|.$$

*The global sensitivity of $f$ is defined as*

$$\mathsf{GS}_f = \sup_{x' \in \mathcal{X}^n} \mathsf{LS}_f(x') = \sup_{x',x'' \in \mathcal{X}^n : d(x'',x') \leq 1} |f(x'') - f(x')|.$$

This allows us to define the smooth sensitivity.

**Definition 14** (Smooth Sensitivity [NRS07]). *Let $f : \mathcal{X}^n \to \mathbb{R}$ and $x \in \mathcal{X}^n$. For $t > 0$, we define the $t$-smoothed sensitivity of $f$ at $x$ as*

$$\mathsf{S}_f^x(t) = \max_{k \geq 0} e^{-tk} \mathsf{LS}_f^k(x) = \sup_{x',x'' \in \mathcal{X}^n : d(x',x'') \leq 1} e^{-td(x,x')} |f(x'') - f(x')|,$$

*where $d(x',x) = |\{i \in [n] : x'_i \neq x_i\}|$ is the number of entries on which $x$ and $x'$ differ.*

The smooth sensitivity has two key properties: First, it is an upper bound on the local sensitivity. Second, it has multiplicative sensitivity bounded by $t$. Indeed, it can be shown that it is the smallest function with these two properties:

**Lemma 15** ([NRS07]). *Let $f, g : \mathcal{X}^n \to \mathbb{R}$. Suppose that, for all $x, x' \in \mathcal{X}^n$ differing in a single entry,*

$$g(x) \geq \mathsf{LS}_f(x) \quad and \quad g(x) \leq e^t g(x').$$

*Then $g(x) \geq \mathsf{S}_f^t(x)$.*

For our applications, we could replace the smooth sensitivity with any function $g$ satisfying the conditions of the above lemma. This is useful if, for example, computing the smooth sensitivity exactly is computationally expensive and we instead wish to use an approximation. Nissim, Raskhodnikova, and Smith [NRS07] show, for instance, that an upper bound on the smooth sensitivity of the median can be computed in sublinear time.

## 2.3 Subgaussian Distributions

We study the class of subgaussian distributions.

**Definition 16.** *A distribution $\mathcal{D}$ on $\mathbb{R}$ is $\sigma$-subgaussian if*

$$\exists \mu \in \mathbb{R} \ \forall t \in \mathbb{R} \quad \mathop{\mathbb{E}}_{X \leftarrow \mathcal{D}} \left[ e^{t(X-\mu)} \right] \leq e^{t^2 \sigma^2 / 2}.$$

*We say that a distribution is subgaussian if it is $\sigma$-subgaussian for some finite $\sigma$.*

Note that $N(\mu, \sigma^2)$, a Gaussian with variance $\sigma^2$, is $\sigma$-subgaussian. Furthermore, any distribution supported on the interval $[a, b]$ is $\frac{b-a}{2}$-subgaussian.

If $\mathcal{D}$ is $\sigma$-subgaussian and has mean zero, then $\mathop{\mathbb{E}}_{X \leftarrow \mathcal{D}}[X^2] \leq \sigma^2$ and, for all $\lambda > 0$,

$$\mathop{\mathbb{P}}_{X \leftarrow \mathcal{D}}[X \geq \lambda] = \mathop{\mathbb{E}}_{X \leftarrow \mathcal{D}}[\mathbb{I}[X - \lambda \geq 0]] \leq \mathop{\mathbb{E}}_{X \leftarrow \mathcal{D}}\left[ e^{t(X-\lambda)} \right] = e^{t^2 \sigma^2 / 2 - t\lambda} = e^{-\lambda^2 / 2\sigma^2},$$

where the final equality follows from setting $t = \lambda / \sigma^2$.

# 3 Smooth Sensitivity Noise Distributions

## 3.1 Laplace Log-Normal

**Definition 17.** *Let $X$ and $Y$ be independent random variables with $X$ a standard Laplace (density $e^{-|x|}/2$) and $Y$ a standard Gaussian (density $e^{-y^2/2}/\sqrt{2\pi}$). Let $\sigma > 0$ and $Z = X \cdot e^{\sigma Y}$. The distribution of $Z$ is denoted $\mathsf{LLN}(\sigma)$.*

Note that $\mathsf{LLN}(\sigma)$ is a symmetric distribution. It has mean 0 and variance $2e^{2\sigma^2}$. More generally, for all $p > 0$,

$$\mathop{\mathbb{E}}_{Z \leftarrow \mathsf{LLN}(\sigma)}[|Z|^p] = \Gamma(p+1) \cdot e^{\sigma^2 p^2 / 2},$$

where $\Gamma$ is the gamma function satisfying $\Gamma(p+1) = p!$ if $p$ is an integer.

**Theorem 18.** *Let $Z \leftarrow \mathsf{LLN}(\sigma)$ and $s, t \in \mathbb{R}$. Then, for all $\alpha \in (1, \infty)$,*

$$\left. \begin{array}{l} \mathrm{D}_\alpha\left(Z \big\| e^t Z + s\right) \\ \mathrm{D}_\alpha\left(e^t Z + s \big\| Z\right) \end{array} \right\} \leq \frac{\alpha}{2} \cdot \left( \frac{|t|}{|\sigma|} + e^{\frac{3}{2}\sigma^2} \cdot |s| \right)^2 \leq \alpha \cdot \left( \frac{t^2}{\sigma^2} + e^{3\sigma^2} s^2 \right).$$

**Lemma 19.** *Let $Z \leftarrow \mathsf{LLN}(\sigma)$ for $\sigma > 0$. Let $t \in \mathbb{R}$ and $\alpha \in (1, \infty)$. Then*

$$\mathrm{D}_\alpha\left(Z \big\| e^t Z\right) \leq \frac{\alpha t^2}{2\sigma^2}.$$

*Proof.* Let $X$ be a standard Laplace random variable and $Y$ an independent standard Gaussian random variable. Let $Z = X e^{\sigma Y} \sim \mathsf{LLN}(\sigma)$. By the quasi-convexity and postprocessing properties of Rényi divergence [BS16, Lem. 2.2], we have

$$\mathrm{D}_\alpha\left(Z \big\| e^t Z\right) = \mathrm{D}_\alpha\left(X e^{\sigma Y} \big\| X e^{\sigma Y + t}\right) \leq \sup_x \mathrm{D}_\alpha\left(x e^{\sigma Y} \big\| x e^{\sigma Y + t}\right) \leq \mathrm{D}_\alpha\left(\sigma Y \big\| \sigma Y + t\right).$$

Finally, we can calculate that $\mathrm{D}_\alpha\left(\sigma Y \| \sigma Y + t\right) = \frac{\alpha t^2}{2\sigma^2}$ [BS16, Lem. 2.4]. $\square$

**Lemma 20.** *Let $Z \leftarrow \mathsf{LLN}(\sigma)$ for $\sigma > 0$. Let $s \in \mathbb{R}$ and $\alpha \in (1, \infty)$. Then*

$$\mathrm{D}_\alpha\left(Z\|Z+s\right) \leq \min\left\{\frac{1}{2}e^{3\sigma^2}s^2\alpha, e^{\frac{3}{2}\sigma^2}s\right\}.$$

*Proof.* Let $Z \leftarrow \mathsf{LLN}(\sigma)$ for some $\sigma > 0$. First we compute the probability density function of $Z$: Let $z > 0$. (Note that $Z$ is symmetric about the origin, so $f(-z) = f(z)$.)

$$
\begin{aligned}
f(z) &= \frac{\mathrm{d}}{\mathrm{d}z} \mathop{\mathbb{E}}_{Y \leftarrow \mathcal{N}(0,1)}\left[\mathop{\mathbb{P}}_{X \leftarrow \mathsf{Lap}(1)}\left[X \cdot e^{\sigma Y} \leq z\right]\right] \\
&= \mathop{\mathbb{E}}_{Y \leftarrow \mathcal{N}(0,1)}\left[\frac{\mathrm{d}}{\mathrm{d}z} \mathop{\mathbb{P}}_{X \leftarrow \mathsf{Lap}(1)}\left[X \leq z \cdot e^{-\sigma Y}\right]\right] \\
&= \mathop{\mathbb{E}}_{Y \leftarrow \mathcal{N}(0,1)}\left[\frac{1}{2}e^{-\left|z \cdot e^{-\sigma Y}\right|} \cdot e^{-\sigma Y}\right] \\
&= \int_{\mathbb{R}} \frac{1}{\sqrt{2\pi}} e^{-y^2/2} \cdot \frac{1}{2} e^{-\left|z \cdot e^{-\sigma y}\right|} \cdot e^{-\sigma y} \mathrm{d}y \\
&= \frac{1}{2\sqrt{2\pi}} \int_{\mathbb{R}} e^{-y^2/2 - \sigma y} e^{-z e^{-\sigma y}} \mathrm{d}y \\
&= \frac{1}{2\sqrt{2\pi}} \int_{\mathbb{R}} e^{-(y-\sigma)^2/2 - \sigma(y-\sigma)} e^{-z e^{-\sigma(y-\sigma)}} \mathrm{d}y \\
&= \frac{1}{2\sqrt{2\pi}} \int_{\mathbb{R}} e^{-y^2/2 + \sigma^2/2} e^{-z e^{\sigma^2} e^{-\sigma y}} \mathrm{d}y \\
&= \frac{e^{\sigma^2/2}}{2} \mathop{\mathbb{E}}_{Y \leftarrow \mathcal{N}(0,1)}\left[\exp\left(-z e^{\sigma^2} e^{\sigma Y}\right)\right].
\end{aligned}
$$

Next we compute the derivative of the density:

$$
\begin{aligned}
f'(z) &= \frac{\mathrm{d}}{\mathrm{d}z} \frac{e^{\sigma^2/2}}{2} \mathop{\mathbb{E}}_{Y \leftarrow \mathcal{N}(0,1)}\left[\exp\left(-z e^{\sigma^2} e^{\sigma Y}\right)\right] \\
&= \frac{e^{\sigma^2/2}}{2} \mathop{\mathbb{E}}_{Y \leftarrow \mathcal{N}(0,1)}\left[\frac{\mathrm{d}}{\mathrm{d}z} \exp\left(-z e^{\sigma^2} e^{\sigma Y}\right)\right] \\
&= \frac{e^{\sigma^2/2}}{2} \mathop{\mathbb{E}}_{Y \leftarrow \mathcal{N}(0,1)}\left[\exp\left(-z e^{\sigma^2} e^{\sigma Y}\right) \cdot (-1) e^{\sigma^2} e^{\sigma Y}\right] \\
&= -e^{\sigma^2} \cdot \frac{e^{\sigma^2/2}}{2\sqrt{2\pi}} \int_{\mathbb{R}} \exp\left(-\frac{y^2}{2} + \sigma y - z e^{\sigma^2} e^{\sigma y}\right) \mathrm{d}y \\
&= -e^{\sigma^2} \cdot \frac{e^{\sigma^2/2}}{2\sqrt{2\pi}} \int_{\mathbb{R}} \exp\left(-\frac{(y+\sigma)^2}{2} + \sigma(y+\sigma) - z e^{\sigma^2} e^{\sigma(y+\sigma)}\right) \mathrm{d}y \\
&= -e^{\sigma^2} \cdot \frac{e^{\sigma^2/2}}{2\sqrt{2\pi}} \int_{\mathbb{R}} \exp\left(-\frac{y^2}{2} + \frac{\sigma^2}{2} - z e^{2\sigma^2} e^{\sigma y}\right) \mathrm{d}y \\
&= -\frac{e^{2\sigma^2}}{2} \mathop{\mathbb{E}}_{Y \leftarrow \mathcal{N}(0,1)}\left[\exp\left(-z e^{2\sigma^2} e^{\sigma Y}\right)\right].
\end{aligned}
$$

Now we can bound the derivative of the log density:

$$
\begin{aligned}
\frac{\mathrm{d}}{\mathrm{d}z} \log f(z) &= \frac{f'(z)}{f(z)} \\
&= \frac{-\frac{e^{2\sigma^2}}{2} \mathop{\mathbb{E}}_{Y \leftarrow \mathcal{N}(0,1)} \left[ \exp\left( -z e^{2\sigma^2} e^{\sigma Y} \right) \right]}{\frac{e^{\sigma^2/2}}{2} \mathop{\mathbb{E}}_{Y \leftarrow \mathcal{N}(0,1)} \left[ \exp\left( -z e^{\sigma^2} e^{\sigma Y} \right) \right]} \\
&= -e^{\frac{3}{2}\sigma^2} \frac{\mathop{\mathbb{E}}_{Y \leftarrow \mathcal{N}(0,1)} \left[ \exp\left( -z e^{\sigma^2} e^{\sigma Y} \right) \cdot \exp\left( -z(e^{\sigma^2} - 1) e^{\sigma^2} e^{\sigma Y} \right) \right]}{\mathop{\mathbb{E}}_{Y \leftarrow \mathcal{N}(0,1)} \left[ \exp\left( -z e^{\sigma^2} e^{\sigma Y} \right) \right]} \\
&\in [-e^{\frac{3}{2}\sigma^2}, 0],
\end{aligned}
$$

since $0 < \exp\left( -z(e^{\sigma^2} - 1) e^{\sigma^2} e^{\sigma Y} \right) \leq 1$. Thus $|\log(f(z)) - \log(f(z+s))| \leq s \cdot e^{\frac{3}{2}\sigma^2}$ for all $s, z \in \mathbb{R}$. Consequently, $\mathrm{D}_\infty\left( Z \| Z + s \right) \leq e^{\frac{3}{2}\sigma^2} \cdot s$ for all $s \in \mathbb{R}$. Note that $\mathrm{D}_\alpha\left( Z \| Z + s \right) \leq \mathrm{D}_\infty\left( Z \| Z + s \right)$ [BS16].

By Proposition 10, $\mathrm{D}_\alpha\left( X \| X + s \right) \leq \frac{1}{2} e^{3\sigma^2} \cdot s^2$ for all $s \in \mathbb{R}$ and all $\alpha \in (1, \infty)$. $\qquad\square$

*Proof of Theorem 18.* We use Lemma 11 to combine Lemmas 19 and 20. Specifically, to bound $\mathrm{D}_\alpha\left( Z \| e^t Z + s \right) = \mathrm{D}_\alpha\left( Z - s \| e^t Z \right)$, we use $\mathrm{D}_\alpha\left( Z - s \| Z \right) = \mathrm{D}_\alpha\left( Z \| Z + s \right)$ and $\mathrm{D}_\alpha\left( Z \| e^t Z \right)$. To bound $\mathrm{D}_\alpha\left( e^t Z + s \| Z \right) = \mathrm{D}_\alpha\left( e^t Z \| Z - s \right)$, we use $\mathrm{D}_\alpha\left( e^t Z \| Z \right) = \mathrm{D}_\alpha\left( Z \| e^{-t} Z \right)$ and $\mathrm{D}_\alpha\left( Z \| Z - s \right)$. $\qquad\square$

### 3.1.1   Optimizing Parameters

Let $f, g : \mathcal{X}^n \to \mathbb{R}$ and $s, t, \sigma > 0$. Suppose, for all neighbouring $x, x' \in \mathcal{X}^n$, we have

$$
|f(x) - f(x')| \leq g(x) \qquad \text{and} \qquad e^{-t} g(x) \leq g(x') \leq e^t g(x).
$$

Define a randomized algorithm $M : \mathcal{X}^n \to \mathbb{R}$ by

$$
M(x) = f(x) + \frac{g(x)}{s} \cdot Z \qquad \text{for} \qquad Z \leftarrow \mathsf{LLN}(\sigma).
$$

Then, by Theorem 18, $M$ is $\frac{1}{2}\varepsilon^2$-CDP for

$$
\varepsilon = \frac{t}{\sigma} + e^{\frac{3}{2}\sigma^2} \cdot s.
$$

Namely, we have

$$
\mathrm{D}_\alpha\left( M(x) \| M(x') \right) = \mathrm{D}_\alpha\left( Z \left\| \frac{f(x') - f(x)}{g(x)} \cdot s + \frac{g(x')}{g(x)} \cdot Z \right. \right).
$$

We also have

$$
\mathbb{E}\left[ (M(x) - f(x))^2 \right] = \frac{g(x)^2}{s^2} 2 e^{2\sigma^2}.
$$

Our goal is to minimize this error for a fixed $\varepsilon$ and $t$ by setting $s$ and $\sigma$. Clearly we should set $s = e^{-\frac{3}{2}\sigma^2}(\varepsilon - t/\sigma)$ to make the constraint tight. Thus our goal is to pick $\sigma > 0$ to minimize

$$\mathbb{E}\left[(M(x) - f(x))^2\right] = \frac{g(x)^2}{e^{-3\sigma^2}(\varepsilon - t/\sigma)^2} 2e^{2\sigma^2} = \frac{2g(x)^2}{e^{-5\sigma^2}(\varepsilon - t/\sigma)^2}.$$

Note that we need $\varepsilon\sigma > t$. Now we maximize the denominator:

$$h(\sigma) = e^{-5\sigma^2}(\varepsilon - t/\sigma)^2$$
$$h'(\sigma) = e^{-5\sigma^2}(-10\sigma)(\varepsilon - t/\sigma)^2 + e^{-5\sigma^2}2(\varepsilon - t/\sigma)(+t/\sigma^2)$$
$$= -e^{-5\sigma^2}\sigma^{-3}\left(10\sigma^2(\varepsilon\sigma - t) - 2t\right)\underbrace{(\varepsilon\sigma - t)}_{>0}$$

$$h'(\sigma) = 0 \iff 10\sigma^2(\varepsilon\sigma - t) - 2t = 0$$
$$\iff \frac{5\varepsilon}{t}\cdot\sigma^3 - 5\sigma^2 - 1 = 0$$

$$\frac{\mathrm{d}}{\mathrm{d}\sigma}\left(\frac{5\varepsilon}{t}\cdot\sigma^3 - 5\sigma^2 - 1\right) = \frac{15\varepsilon}{t}\cdot\sigma^2 - 10\sigma = 5\sigma\left(\frac{3\varepsilon}{t}\cdot\sigma - 2\right) > 5\sigma > 0.$$

The inequality in the underbrace follows from the fact that we restrict ourselves to $\varepsilon\sigma > t$ (as otherwise the problem is infeasible). So to minimize variance, we must pick $\sigma$ such that $5\frac{\varepsilon}{t}\sigma^3 - 5\sigma^2 - 1 = 0$. Since the derivative of the right hand side of this with respect to $\sigma$ is strictly positive, the cubic equation has exactly one real root.

Substituting $\sigma = t/\varepsilon$ into the cubic yields a strictly negative number. Thus the solution is strictly larger than this (as required). Substituting $\sigma = \max\{2t/\varepsilon, 1/2\}$ yields a strictly positive value for the cubic, yielding an upper bound on the solution. This means we can find the solution numerically using binary search.

## 3.2 Uniform Log-Normal

**Definition 21.** *Let $X$ and $Y$ be independent random variables, where $X$ is uniform on $[-1, 1]$ and $Y$ is a standard Gaussian. Let $\sigma > 0$ and $Z = X \cdot e^{\sigma Y}$. The distribution of $Z$ is denoted $\mathsf{ULN}(\sigma)$.*

Note that $\mathsf{ULN}(\sigma)$ is a symmetric distribution. It has mean 0 and variance $\frac{1}{3}e^{2\sigma^2}$. More generally, for all $p > 0$,

$$\mathbb{E}_{Z \leftarrow \mathsf{ULN}(\sigma)}[|Z|^p] = \frac{1}{p+1}e^{\sigma^2 p^2/2}.$$

**Theorem 22.** *Let $Z \leftarrow \mathsf{ULN}(\sigma)$ with $\sigma \geq \sqrt{2}$ and $s, t \in \mathbb{R}$. Then, for all $\alpha \in (1, \infty)$,*

$$\left.\begin{array}{l}\mathrm{D}_\alpha\left(Z\|e^t Z + s\right) \\ \mathrm{D}_\alpha\left(e^t Z + s\|Z\right)\end{array}\right\} \leq \frac{\alpha}{2}\cdot\left(\frac{|t|}{|\sigma|} + e^{\frac{3}{2}\sigma^2}\sqrt{\frac{2}{\pi\sigma^2}}\cdot|s|\right)^2 \leq \alpha\cdot\left(\frac{t^2}{\sigma^2} + e^{3\sigma^2}\frac{2}{\pi\sigma^2}s^2\right).$$

The proof of Theorem 22 closely follows that of Theorem 18

**Lemma 23.** *Let $Z \leftarrow \mathsf{ULN}(\sigma)$ for $\sigma > 0$. Let $t \in \mathbb{R}$ and $\alpha \in (1, \infty)$. Then*

$$\mathrm{D}_\alpha\left(Z \| e^t Z\right) \leq \frac{\alpha t^2}{2\sigma^2}.$$

The proof is identical to that of Lemma 19.

**Lemma 24.** *Let $Z \leftarrow \mathsf{ULN}(\sigma)$ for $\sigma^2 \geq 2$. Let $s \in \mathbb{R}$ and $\alpha \in (1, \infty)$. Then*

$$\mathrm{D}_\alpha\left(Z \| Z + s\right) \leq \min\left\{\frac{e^{3\sigma^2}}{\pi\sigma^2} s^2 \alpha, \frac{e^{\frac{3}{2}\sigma^2}}{\sigma\sqrt{\pi/2}} s\right\}.$$

*Proof.* We compute the probability density $f_Z$ of $Z \leftarrow \mathsf{ULN}(\sigma)$. Since the distribution is symmetric we can restrict our attention to $z > 0$.

$$
\begin{aligned}
f_Z(z) &= \frac{\mathrm{d}}{\mathrm{d}z} \mathbb{P}\left[Z \leq z\right] \\
&= \frac{\mathrm{d}}{\mathrm{d}z} \mathbb{E}_Y\left[\mathbb{P}_X\left[X \cdot e^{\sigma Y} \leq z\right]\right] \\
&= \mathbb{E}_Y\left[\frac{\mathrm{d}}{\mathrm{d}z} \mathbb{P}_X\left[X \leq z e^{-\sigma Y}\right]\right] \\
&= \mathbb{E}_Y\left[\frac{1}{2}\mathbb{I}[z e^{-\sigma Y} \leq 1] \cdot e^{-\sigma Y}\right] \\
&= \frac{1}{\sqrt{2\pi}} \int_{-\infty}^{\infty} \frac{1}{2}\mathbb{I}[z e^{-\sigma y} \leq 1] \cdot e^{-\sigma y - y^2/2} \mathrm{d}y \\
&= \frac{1}{\sqrt{2\pi}} \int_{-\infty}^{\infty} \frac{1}{2}\mathbb{I}[z e^{-\sigma(y-\sigma)} \leq 1] \cdot e^{-\sigma(y-\sigma)-(y-\sigma)^2/2} \mathrm{d}y \\
&= \frac{1}{\sqrt{2\pi}} \int_{-\infty}^{\infty} \frac{1}{2}\mathbb{I}\left[y \geq \sigma + \frac{1}{\sigma}\log z\right] \cdot e^{\sigma^2/2 - y^2/2} \mathrm{d}y \\
&= \frac{e^{\sigma^2/2}}{2\sqrt{2\pi}} \int_{\sigma+\frac{1}{\sigma}\log z}^{\infty} e^{-y^2/2} \mathrm{d}y \\
&= \frac{e^{\sigma^2/2}}{2\sqrt{2\pi}} \int_{g(z)}^{\infty} e^{-y^2/2} \mathrm{d}y \\
&= \frac{e^{\sigma^2/2}}{2} \mathbb{P}_{Y \leftarrow \mathcal{N}(0,1)}\left[Y \geq g(z)\right],
\end{aligned}
$$

where $g(z) := \sigma + \frac{1}{\sigma}\log z$. We can also calculate the derivative of the density from the

fundamental theorem of calculus:

$$f'_Z(z) = \frac{\mathrm{d}}{\mathrm{d}z} \frac{e^{\sigma^2/2}}{2\sqrt{2\pi}} \int_{g(z)}^{\infty} e^{-y^2/2} \mathrm{d}y$$

$$= -\frac{e^{\sigma^2/2}}{2\sqrt{2\pi}} e^{-g(z)^2/2} \cdot g'(z)$$

$$= -\frac{e^{\sigma^2/2}}{2\sqrt{2\pi}} e^{-g(z)^2/2} \cdot \frac{1}{\sigma z}.$$

Clearly $f'_Z(z) < 0$ (for $z > 0$). This shows that the distribution is unimodal. We also have a second derivative:

$$f''_Z(z) = \frac{\mathrm{d}}{\mathrm{d}z} \left( -\frac{e^{\sigma^2/2}}{2\sqrt{2\pi}} e^{-g(z)^2/2} \cdot \frac{1}{\sigma z} \right)$$

$$= -\frac{e^{\sigma^2/2}}{2\sqrt{2\pi}} e^{-g(z)^2/2} \left( -g(z)g'(z) \cdot \frac{1}{\sigma z} - \frac{1}{\sigma z^2} \right)$$

$$= \frac{e^{\sigma^2/2}}{2\sqrt{2\pi}\sigma} \frac{e^{-g(z)^2/2}}{z^2} \left( \frac{\log z}{\sigma^2} + 2 \right).$$

The second derivative is negative if $z < e^{-2\sigma^2}$ and positive if $z > e^{-2\sigma^2}$. Noting that the first derivative is always negative, this shows that the maximum magnitude of the first derivative is attained at $z = e^{-2\sigma^2}$. That is, for all $z > 0$, we have

$$-\frac{e^{2\sigma^2}}{2\sqrt{2\pi}\sigma} \leq f'_Z(z) \leq 0.$$

Our goal is to bound

$$\frac{\mathrm{d}}{\mathrm{d}z} \log f_Z(z) = \frac{f'_Z(z)}{f_Z(z)} = \frac{-1}{\sigma} \cdot \frac{\frac{1}{z} e^{-g(z)^2/2}}{\int_{g(z)}^{\infty} e^{-y^2/2} \mathrm{d}y}.$$

We already have a uniform upper bound on the numerator: for all $z > 0$,

$$0 \leq \frac{1}{z} e^{-g(z)^2/2} \leq e^{\frac{3}{2}\sigma^2}.$$

Next we prove lower bounds on the denominator. Firstly, for any $a \leq 0$,

$$\int_a^{\infty} e^{-y^2/2} \mathrm{d}y \geq \int_0^{\infty} e^{-y^2/2} \mathrm{d}y = \sqrt{\frac{\pi}{2}}.$$

Thus, for $z \leq e^{-\sigma^2}$, we have $g(z) \leq 0$ and

$$\left| \frac{\mathrm{d}}{\mathrm{d}z} \log f_Z(z) \right| \leq \frac{e^{\frac{3}{2}\sigma^2}}{\sigma\sqrt{\pi/2}}.$$

For $a, b \geq 0$,

$$\int_a^\infty e^{-y^2/2}\mathrm{d}y \geq \int_a^{a+b} e^{-y^2/2}\mathrm{d}y$$

$$= \int_0^b e^{-(y+a)^2/2}\mathrm{d}y$$

$$\geq \int_0^b \exp\left(\frac{-1}{2}\left(\frac{y}{b}\cdot(a+b)^2 + (1-\frac{y}{b})a^2\right)\right)\mathrm{d}y$$

$$= \int_0^b \exp\left(\frac{-1}{2}\left(a^2 + \frac{y}{b}(2ab+b^2)\right)\right)\mathrm{d}y$$

$$= e^{-a^2/2}\int_0^b e^{-(a+b/2)y}\mathrm{d}y$$

$$= e^{-a^2/2}\int_0^b \left(\frac{\mathrm{d}}{\mathrm{d}y}\frac{e^{-(a+b/2)y}}{-(a+b/2)}\right)\mathrm{d}y$$

$$= e^{-a^2/2}\cdot\frac{1-e^{-(a+b/2)b}}{a+b/2},$$

where the second inequality uses the fact that $x \mapsto -x^2$ is concave. Hence, for any $z > e^{-\sigma^2}$ and $b > 0$, we have $g(z) > 0$ and

$$\left|\frac{\mathrm{d}}{\mathrm{d}z}\log f_Z(z)\right| = \frac{1}{\sigma}\cdot\frac{\frac{1}{z}e^{-g(z)^2/2}}{\int_{g(z)}^\infty e^{-y^2/2}\mathrm{d}y}$$

$$\leq \frac{1}{\sigma}\cdot\frac{\frac{1}{z}e^{-g(z)^2/2}}{e^{-g(z)^2/2}\cdot\frac{1-e^{-(g(z)+b/2)b}}{g(z)+b/2}}$$

$$= \frac{1}{\sigma}\cdot\frac{g(z)+b/2}{z}\frac{1}{1-e^{-(g(z)+b/2)b}}$$

$$= \left(\frac{1+b/2\sigma}{z} + \frac{\log z}{\sigma^2 z}\right)\frac{1}{1-e^{-(g(z)+b/2)b}}$$

$$\leq \left(\frac{1+b/2\sigma}{e^{-\sigma^2}} + \frac{1}{\sigma^2 e}\right)\frac{1}{1-e^{-b^2/2}},$$

since $\log z \leq z/e$. It only remains to set $b$. Setting $b = 2$ gives, for $z > e^{-\sigma^2}$ and $\sigma \geq \sqrt{2}$,

$$\left|\frac{\mathrm{d}}{\mathrm{d}z}\log f_Z(z)\right| \leq \left(\frac{1+1/\sigma}{e^{-\sigma^2}} + \frac{1}{\sigma^2 e}\right)\frac{1}{1-e^{-2}} \leq \frac{e^{\frac{3}{2}\sigma^2}}{\sigma\sqrt{\pi/2}}.$$

To justify the final inequality above, note that it is equivalent to

$$\frac{\sqrt{\pi/2}}{1+e^{-2}}\left(\frac{1+\sigma}{e^{\sigma^2/2}} + \frac{1}{\sigma e^{1+\sigma^2/2}}\right) \leq 1,$$

which can be verified numerically to hold for $\sigma = \sqrt{2}$. It is also easy to show that the left hand side is a decreasing function of $\sigma$ for $\sigma \geq 1$, which establishes it for all $\sigma \geq \sqrt{2}$. This bound on $\left| \frac{d}{dz} \log f_Z(z) \right|$ entails a pure differential privacy guarantee for additive distortion, which by Proposition 10 gives the concentrated differential privacy guarantee. $\square$

## 3.3 Arsinh-Normal

**Theorem 25.** *Let $Y$ be a standard Gaussian and $\sigma > 0$. Let $X = \frac{1}{\sigma} \sinh(\sigma Y)$. Let $s, t \in \mathbb{R}$. Then, for all $\alpha \in (1, \infty)$,*

$$\left. \begin{array}{l} \mathrm{D}_\alpha \left( X \| e^t X + s \right) \\ \mathrm{D}_\alpha \left( e^t X + s \| X \right) \end{array} \right\} \leq \frac{\alpha}{2} \cdot \left( \sqrt{|t| \cdot \left( \frac{|t|}{\sigma^2} + \frac{1}{\sigma} + 2 \right)} + |s| \cdot \left( \frac{2}{3\sigma} + \frac{\sigma}{2} \right) \right)^2.$$

We can calculate that $\mathbb{E}_{Y \leftarrow N(0,1)} \left[ \left( \frac{1}{\sigma^2} \sinh(\sigma Y) \right)^2 \right] = \frac{e^{2\sigma^2} - 1}{2\sigma^2}$. In particular, setting $\sigma = 2/\sqrt{3}$ and requiring $|t| \leq 1/2$, yields

$$\left. \begin{array}{l} \mathrm{D}_\alpha \left( X \| e^t X + s \right) \\ \mathrm{D}_\alpha \left( e^t X + s \| X \right) \end{array} \right\} \leq \frac{\alpha}{2} \cdot \left( 1.81 \sqrt{|t|} + 1.16 |s| \right)^2$$

and $\mathbb{E}\left[ X^2 \right] \leq 5.03$.

**Lemma 26.** *Let $Y$ be a standard Gaussian and $\sigma > 0$. Let $X = \frac{1}{\sigma} \sinh(\sigma Y)$. Then the density of $X$ is given by*

$$f_X(x) = \frac{1}{\sqrt{2\pi}} e^{-(\mathrm{arsinh}(\sigma x))^2 / 2\sigma^2} \cdot \frac{1}{\sqrt{1 + (\sigma x)^2}}.$$

This follows from the change-of-variables lemma

**Lemma 27.** *Let $Y$ be a distribution on $\mathbb{R}$ with density $f_Y$. Let $g : \mathbb{R} \to \mathbb{R}$ be an increasing and differentiable function. Let $X = g^{-1}(Y)$. Then the density of $X$ is given by*

$$f_X(x) = f_Y(g(x)) \cdot g'(x).$$

**Lemma 28.** *Let $Y$ be a standard Gaussian and $\sigma > 0$. Let $X = \frac{1}{\sigma} \sinh(\sigma Y)$. Then*

$$\mathrm{D}_\infty \left( X \| X + s \right) \leq |s| \cdot \left( \frac{2}{3\sigma} + \frac{\sigma}{2} \right)$$

*for all $s \in \mathbb{R}$.*

Note that setting $\sigma = 2/\sqrt{3} \approx 1.15$ yields $\mathrm{D}_\infty \left( X \| X + s \right) \leq |s| \cdot \frac{2}{\sqrt{3}}$ for all $s \in \mathbb{R}$.

*Proof.* To prove this we simply need to show that

$$\left|\frac{\mathrm{d}}{\mathrm{d}x}\log(f_X(x))\right| = \left|\frac{f_X'(x)}{f_X(x)}\right| \le \frac{2}{3\sigma} + \frac{\sigma}{2}$$

for all $x \in \mathbb{R}$, where $f_X$ is the density of $X$ given by

$$f_X(x) = \frac{1}{\sqrt{2\pi}}e^{-(\operatorname{arsinh}(\sigma x))^2/2\sigma^2} \cdot \frac{1}{\sqrt{1+(\sigma x)^2}}.$$

We have

$$\begin{aligned}
f_X'(x) &= \frac{1}{\sqrt{2\pi}}e^{-(\operatorname{arsinh}(\sigma x))^2/2\sigma^2} \cdot \frac{-\operatorname{arsinh}(\sigma x)}{\sigma^2} \cdot \frac{\sigma}{\sqrt{1+(\sigma x)^2}} \cdot \frac{1}{\sqrt{1+(\sigma x)^2}} \\
&\quad + \frac{1}{\sqrt{2\pi}}e^{-(\operatorname{arsinh}(\sigma x))^2/2\sigma^2} \cdot \frac{-1}{2(1+(\sigma x)^2)^{3/2}} \cdot 2\sigma^2 x \\
&= -f_X(x) \cdot \left(\frac{\operatorname{arsinh}(\sigma x)}{\sigma\sqrt{1+(\sigma x)^2}} + \frac{\sigma^2 x}{1+(\sigma x)^2}\right) \\
&= -f_X(x) \cdot \left(\frac{1}{\sigma} \cdot \frac{\operatorname{arsinh}(u)}{\sqrt{1+u^2}} + \sigma \cdot \frac{u}{1+u^2}\right),
\end{aligned}$$

where $u = \sigma x$. We have $\left|\frac{u}{1+u^2}\right| \le \frac{1}{2}$ and $\left|\frac{\operatorname{arsinh}(u)}{\sqrt{1+u^2}}\right| \le \frac{2}{3}$ for all $u \in \mathbb{R}$. This yields the result. $\qquad\square$

**Lemma 29.** *Let $Y$ be a standard Gaussian and $\sigma > 0$. Let $X = \frac{1}{\sigma}\sinh(\sigma Y)$. Then*

$$D_\alpha\left(X\|e^t X\right) \le \alpha \cdot \frac{t^2}{2\sigma^2} + \frac{|t|}{2\sigma} + \max\{0, t\} \le \alpha|t|\left(\frac{|t|}{2\sigma^2} + \frac{1}{2\sigma} + 1\right)$$

*for all $t \in \mathbb{R}$ and all $\alpha \in (1, \infty)$.*

*Proof.* Fix $t \in \mathbb{R}$ and all $\alpha \in (1, \infty)$. We have

$$D_\alpha\left(X\|e^t X\right) = D_\alpha\left(\frac{1}{\sigma}\sinh(\sigma Y)\,\middle\|\,\frac{e^t}{\sigma}\sinh(\sigma Y)\right) = D_\alpha\left(Y\,\middle\|\,\frac{1}{\sigma}\operatorname{arsinh}\left(e^t\sinh(\sigma Y)\right)\right) = D_\alpha\left(Y\|g^{-1}(Y)\right),$$

where $g(x) = \frac{1}{\sigma}\operatorname{arsinh}(e^{-t}\sinh(\sigma x))$. By Lemma 27, the density of $g^{-1}(Y)$ is given by

$f_{g^{-1}(Y)}(y) = f_Y(g(y)) \cdot g'(y)$. Hence

$$e^{(\alpha-1)\mathrm{D}_\alpha\left(X\|e^t X\right)} = e^{(\alpha-1)\mathrm{D}_\alpha\left(Y\|g^{-1}(Y)\right)}$$

$$= \mathop{\mathbb{E}}_{Y \leftarrow N(0,1)} \left[ \left( \frac{f_Y(Y)}{f_{g^{-1}(Y)}(Y)} \right)^{\alpha-1} \right]$$

$$= \mathop{\mathbb{E}}_{Y \leftarrow N(0,1)} \left[ \left( \frac{f_Y(Y)}{f_Y(g(Y)) \cdot g'(Y)} \right)^{\alpha-1} \right]$$

$$= \mathop{\mathbb{E}}_{Y \leftarrow N(0,1)} \left[ e^{\frac{\alpha-1}{2}(g(Y)^2 - Y^2)} \cdot \frac{1}{(g'(Y))^{\alpha-1}} \right]$$

$$\leq \mathop{\mathbb{E}}_{Y \leftarrow N(0,1)} \left[ e^{\frac{\alpha-1}{2}(g(Y)^2 - Y^2)} \right] \cdot \left( \sup_{y \in \mathbb{R}} \frac{1}{g'(y)} \right)^{\alpha-1}.$$

We now bound these terms one by one.

We have, for all $y \in \mathbb{R}$,

$$g'(y) = \frac{e^{-t} \cosh(\sigma y)}{\sqrt{1 + e^{-2t} \sinh(\sigma y)^2}} > 0.$$

Thus, for all $y \in \mathbb{R}$,

$$\left( \frac{1}{g'(y)} \right)^2 = \frac{1 + e^{-2t} \sinh(\sigma y)^2}{e^{-2t} \cosh(\sigma y)^2}$$

$$= \frac{1 + e^{-2t}(\cosh(\sigma y)^2 - 1)}{e^{-2t} \cosh(\sigma y)^2}$$

$$= 1 + \frac{1 - e^{-2t}}{e^{-2t} \cosh(\sigma y)^2}$$

$$= 1 + \frac{e^{2t} - 1}{\cosh(\sigma y)^2}$$

$$\leq \max\{1, e^{2t}\}.$$

This implies that $\sup_{y \in \mathbb{R}} \frac{1}{g'(y)} \leq \max\{1, e^t\} = e^{\max\{0,t\}}$.

Now we consider $y \in \mathbb{R}$ fixed and let $t \in \mathbb{R}$ vary: Define

$$h(t) = g(y) = \frac{1}{\sigma} \operatorname{arsinh}(e^{-t} \sinh(\sigma y)).$$

Then $h(0) = y$ and

$$h'(t) = \frac{1}{\sigma} \frac{1}{\sqrt{1 + (e^{-t} \sinh(\sigma y))^2}} \cdot e^{-t} \sinh(\sigma y)(-1)$$

$$= \frac{-1}{\sigma} \frac{1}{\sqrt{1 + (e^{-t} \sinh(\sigma y))^{-2}}}$$

$$\in \left[ \frac{-1}{\sigma}, 0 \right].$$

This implies that, for all $y \in \mathbb{R}$,

$$\frac{\min\{0, -t\}}{\sigma} \leq g(y) - y \leq \frac{\max\{0, -t\}}{\sigma}$$

and

$$g(y)^2 - y^2 = (g(y) - y)^2 + (g(y) - y) \cdot 2y \leq \frac{t^2}{\sigma^2} + \frac{2}{\sigma} \max\{0, -ty\}.$$

Now we calculate:

$$\mathbb{E}_{Y \leftarrow N(0,1)} \left[ e^{\frac{\alpha-1}{2}(g(Y)^2 - Y^2)} \right] \leq \mathbb{E}_{Y \leftarrow N(0,1)} \left[ e^{\frac{\alpha-1}{2}(t^2/\sigma^2 + 2\max\{0, -tY\}/\sigma)} \right]$$

$$= e^{(\alpha-1)t^2/2\sigma^2} \cdot \mathbb{E}_{Y \leftarrow N(0,1)} \left[ e^{(\alpha-1)\max\{0, |t|Y\}/\sigma} \right]$$

$$(v := (\alpha-1)|t|/\sigma \geq 0) \quad = e^{(\alpha-1)t^2/2\sigma^2} \cdot \left( \frac{1}{2} + \int_0^\infty e^{vy} \cdot \frac{e^{-y^2/2}}{\sqrt{2\pi}} \, dy \right)$$

$$(y \mapsto \hat{y} + v) \quad = e^{(\alpha-1)t^2/2\sigma^2} \cdot \left( \frac{1}{2} + \int_{-v}^\infty e^{v(\hat{y}+v)} \cdot \frac{e^{-(\hat{y}+v)^2/2}}{\sqrt{2\pi}} \, d\hat{y} \right)$$

$$= e^{(\alpha-1)t^2/2\sigma^2} \cdot \left( \frac{1}{2} + \int_{-v}^\infty e^{v^2/2} \cdot \frac{e^{-\hat{y}^2/2}}{\sqrt{2\pi}} \, d\hat{y} \right)$$

$$= e^{(\alpha-1)t^2/2\sigma^2} \cdot \left( \frac{1}{2} + e^{v^2/2} \cdot \mathbb{P}_{Y \leftarrow N(0,1)} [Y \geq -v] \right)$$

$$\leq e^{(\alpha-1)t^2/2\sigma^2} \cdot \left( \frac{1}{2} + e^{v^2/2} \cdot \left( \frac{1}{2} + \frac{v}{\sqrt{2\pi}} \right) \right)$$

$$\leq e^{(\alpha-1)t^2/2\sigma^2} \cdot e^{v^2/2 + v/2}$$

$$= \exp \left( (\alpha-1) \left( \alpha \cdot \frac{t^2}{2\sigma^2} + \frac{|t|}{2\sigma} \right) \right).$$

Now we complete the calculation:

$$e^{(\alpha-1)\mathrm{D}_\alpha(X \| e^t X)} \leq \mathbb{E}_{Y \leftarrow N(0,1)} \left[ e^{\frac{\alpha-1}{2}(g(Y)^2 - Y^2)} \right] \cdot \left( \sup_{y \in \mathbb{R}} \frac{1}{g'(y)} \right)^{\alpha-1}$$

$$\leq e^{(\alpha-1)\left( \alpha t^2/2\sigma^2 + |t|/2\sigma \right)} \cdot e^{(\alpha-1)\max\{0,t\}}.$$

This implies the result:

$$\mathrm{D}_\alpha \left( X \| e^t X \right) \leq \alpha \cdot \frac{t^2}{2\sigma^2} + \frac{|t|}{2\sigma} + \max\{0, t\}.$$

$\square$

## 3.4 Student's T

Nissim, Raskhodnikova, and Smith [NRS07] showed that distributions with density $\propto \frac{1}{1+|z|^\gamma}$ for any constant $\gamma > 0$ can be scaled to smooth sensitivity to guarantee pure differential privacy. Here we show that the same is true for the family of Student's T distributions, which are defined similarly and have a number of uses in statistics.

**Definition 30.** *The student's T distribution with $d > 0$ degrees of freedom is denoted $\mathsf{T}(d)$ and has a probability density of*

$$f_{\mathsf{T}(d)}(x) = \frac{\Gamma\left(\frac{d+1}{2}\right)}{\sqrt{\pi d}\,\Gamma\left(\frac{d}{2}\right)} \left(\frac{1}{1+\frac{x^2}{d}}\right)^{\frac{d+1}{2}}$$

The Cauchy distribution corresponds to $\mathsf{T}(1)$. The T distribution is centered at zero and has variance $\frac{d}{d-2}$ for $d > 2$. For integral $d \geq 1$ we can sample $Z \leftarrow \mathsf{T}(d)$ by letting $Z = \frac{X_0}{\sqrt{\frac{1}{d}\sum_{i=1}^{d} X_i^2}}$, where $X_0, X_1, \cdots, X_d$ are independent standard Gaussians.

**Theorem 31.** *Let $Z \leftarrow \mathsf{T}(d)$ with $d > 0$ and $s, t \in \mathbb{R}$. Then, for all $\alpha \in (1, \infty)$,*

$$\left.\begin{array}{l} \mathrm{D}_\alpha\left(Z \| e^t Z + s\right) \\ \mathrm{D}_\alpha\left(e^t Z + s \| Z\right) \end{array}\right\} \leq \min\left\{|t| \cdot (d+1) + |s| \cdot \frac{d+1}{2\sqrt{d}}, \frac{\alpha}{2}\left(|t| \cdot (d+1) + |s| \cdot \frac{d+1}{2\sqrt{d}}\right)^2\right\}.$$

*Proof.* Let $Z \leftarrow \mathsf{T}(d)$. We handle the multiplicative distortion first and then the additive distortion. Define

$$g(t) = \log\left(\frac{f_T(e^t x)}{f_T(x)}\right) = \frac{d+1}{2}\left(\log(d+x^2) - \log(d+e^{2t}x^2)\right).$$

Then

$$g'(t) = -\frac{d+1}{2}\frac{2e^{2t}x^2}{d+e^{2t}x^2}$$

and

$$|g'(t)| \leq d+1.$$

Thus $\mathrm{D}_\infty\left(Z \| e^t Z\right) \leq |t| \cdot (d+1)$. Next we handle the additive distortion. Define

$$h(s) = \log\left(\frac{f_T(x+s)}{f_T(x)}\right) = \frac{d+1}{2}\left(\log(d+x^2) - \log(d+(x+s)^2)\right).$$

Then

$$h'(s) = -\frac{d+1}{2}\frac{2(x+s)}{d+(x+s)^2} = -\frac{d+1}{d/y+y}$$

for $y = x + s$. The magnitude is maximized for $y = \sqrt{d}$, yielding the bound

$$|h'(s)| \leq \frac{d+1}{2\sqrt{d}}.$$

Thus $\mathrm{D}_\infty\left(Z \| Z + s\right) \leq |s| \cdot \frac{d+1}{2\sqrt{d}}$. Combining the multiplicative and additive bounds via a triangle-like inequality for max divergence and Proposition 10 yields the result. □

## 3.5 Gaussian & tCDP

For comparison, we consider the Normal distribution under tCDP [BDRS18] – a relaxation of CDP. First we state the definition of tCDP.

**Definition 32** (Truncated CDP [BDRS18])**.** *A randomized algorithm $M : \mathcal{X}^n \to \mathcal{Y}$ is $\left(\frac{1}{2}\varepsilon^2, \omega\right)$-truncated CDP ($\left(\frac{1}{2}\varepsilon^2, \omega\right)$-tCDP) if, for all $x, x' \in \mathcal{X}^n$ differing in a single entry,*

$$\sup_{\alpha \in (1,\omega)} \frac{1}{\alpha} \mathrm{D}_\alpha \left( M(x) \| M(x') \right) \leq \frac{1}{2}\varepsilon^2,$$

*where $\mathrm{D}_\alpha \left( \cdot \| \cdot \right)$ denotes the Rényi divergence of order $\alpha$.*

Gaussian noise with variance $\sigma^2$ scaled to $t$-smooth sensitivity provides $\left( \frac{1}{2\sigma^2(1-\omega(1-e^{-t}))} + \frac{t^2}{4(1-\omega(1-e^{-t}))^2}, \omega \right)$-tCDP for all $\omega < \frac{1}{1-e^{-t}}$.

**Lemma 33** ([BDRS18])**.** *Let $\mu_0, \mu_1, \sigma, t, \gamma, \alpha \in \mathbb{R}$. If $\alpha > 1$ and $\alpha(e^t - 1) + 1 \geq \gamma > 0$, then*

$$\mathrm{D}_\alpha \left( \mathcal{N}(\mu_0, \sigma^2) \| \mathcal{N}(\mu_1, e^t\sigma^2) \right) \leq \alpha \cdot \left( \frac{(\mu_0 - \mu_1)^2}{2\gamma \cdot \sigma^2} + \frac{t^2}{4\gamma^2} \right).$$

## 3.6 Laplace & Approximate DP

As a final comparison, we consider the approximate DP guarantees of Laplace noise. This is a sharpening of the analysis of Nissim, Raskhodnikova, and Smith [NRS07].

**Theorem 34.** *Let $X$ be a standard Laplace random variable and $s, t \in \mathbb{R}$. Then, for all measurable $E \subset \mathbb{R}$,*

$$\mathbb{P}\left[ e^t X + s \in E \right] \leq e^\varepsilon \mathbb{P}\left[ X \in E \right] + \delta$$

*for any $\delta \in (0, e^{-2})$ and*

$$\varepsilon \geq |s| + (e^{|t|} - 1) \log(1/\delta) - |t|.$$

*Proof.* Fix $s, t \in \mathbb{R}$, $c > 0$, and a measurable set $E$. Since $-|x - s|e^{-t} = -|x - s| + (1 - e^{-t})|x - s| \leq -|x| + |s| + (1 - e^{-t})|x - s|$, we have

$$\mathbb{P}\left[ e^t X + s \in E \right] = \int_E \frac{1}{2e^t} e^{-|x-s|e^{-t}} \mathrm{d}x$$

$$= \int_{E \cap [s-c, s+c]} \frac{1}{2e^t} e^{-|x-s|e^{-t}} \mathrm{d}x + \int_{E \setminus [s-c, s+c]} \frac{1}{2e^t} e^{-|x-s|e^{-t}} \mathrm{d}x$$

$$\leq \int_{E \cap [s-c, s+c]} \frac{1}{2e^t} e^{-|x|+|s|+\max\{0, c(1-e^{-t})\}} \mathrm{d}x + \int_{\mathbb{R} \setminus [s-c, s+c]} \frac{1}{2e^t} e^{-|x-s|e^{-t}} \mathrm{d}x$$

$$= e^{|s|-t+\max\{0, c(1-e^{-t})\}} \mathbb{P}\left[ X \in E \cap [s-c, s+c] \right] + \frac{1}{e^t} \int_c^\infty e^{-ye^{-t}} \mathrm{d}y$$

$$\leq e^{|s|-t+\max\{0, c(1-e^{-t})\}} \mathbb{P}\left[ X \in E \right] + e^{-ce^{-t}}.$$

Setting $c = e^t \log(1/\delta)$ yields

$$\mathbb{P}\left[ e^t X + s \in E \right] \leq e^{|s|-t+\max\{0, (e^t-1)\log(1/\delta)\}} \mathbb{P}\left[ X \in E \right] + \delta.$$

$\square$

# 4   Lower Bounds

We provide a lower bound on the tail of any distribution providing concentrated differential privacy for smooth sensitivity. We note that this roughly matches the tail bounds attained by our three distributions.

**Proposition 35.** *Let $s, t, \varepsilon > 0$. Let $Z$ be a real random variable satisfying, for all $\alpha \in (1, \infty)$,*

$$\mathrm{D}_\alpha\left(e^t Z + s \,\middle\|\, Z\right) \le \frac{1}{2}\varepsilon^2 \alpha \quad\text{and}\quad \mathrm{D}_\alpha\left(Z \,\middle\|\, e^t Z - s\right) \le \frac{1}{2}\varepsilon^2 \alpha.$$

*Then, for all $x > 0$,*

$$\mathbb{P}\left[|Z| > x\right] \ge \frac{1}{4} e^{-\varepsilon^2 \left\lceil \frac{1}{t}\log\left(1 + \frac{x}{s}(e^t - 1)\right)\right\rceil^2} = e^{-\Theta\left(\frac{\varepsilon}{t}\log\left(\frac{xt}{s}\right)\right)^2}.$$

*Proof.* We may assume $\mathbb{P}\left[Z \ge 0\right] \ge \frac{1}{2}$. If not, we replace $Z$ with $-Z$ in the argument below.

Fix $x > 0$ and Let $p = \mathbb{P}\left[Z \ge x\right]$. We may assume $p < \frac{1}{2}$, as otherwise the result is trivial.

By postprocessing and group privacy (Lemma 11), for all integers $k \ge 0$ and all $\alpha \in (1, \infty)$,

$$\mathrm{D}_\alpha\left(\mathbb{I}\left[e^t Z + s\frac{e^{kt} - 1}{e^t - 1} \ge x\right] \,\middle\|\, \mathbb{I}[Z \ge x]\right) \le \mathrm{D}_\alpha\left(e^t Z + s\frac{e^{kt} - 1}{e^t - 1} \,\middle\|\, Z\right) \le \frac{1}{2}\varepsilon^2 k^2 \alpha.$$

Note that the indicators above are simply Bernoulli random variables. Let $q_k = \mathbb{P}\left[e^t Z + s\frac{e^{kt}-1}{e^t-1} \ge x\right]$. Then we have $\mathrm{D}_1\left(q_k \,\middle\|\, p\right) := \mathrm{D}_1\left(\mathsf{Bern}(q_k) \,\middle\|\, \mathsf{Bern}(p)\right) \le \frac{1}{2}\varepsilon^2 k^2$.

Suppose $k \ge \frac{1}{t}\log\left(1 + \frac{x}{s}(e^t - 1)\right)$. Then $s\frac{e^{kt}-1}{e^t-1} \ge x$, whence

$$q_k = \mathbb{P}\left[e^t Z + s\frac{e^{kt} - 1}{e^t - 1} \ge x\right] \ge \mathbb{P}\left[Z \ge 0\right] \ge \frac{1}{2}.$$

Thus

$$\frac{1}{2}\varepsilon^2 k^2 \ge \mathrm{D}_1\left(q_k \,\middle\|\, p\right) \ge \mathrm{D}_1\left(\frac{1}{2} \,\middle\|\, p\right) = \frac{1}{2}\log\left(\frac{1}{4p(1-p)}\right).$$

Setting $k = \left\lceil \frac{1}{t}\log\left(1 + \frac{x}{s}(e^t - 1)\right)\right\rceil = \Theta(\log(xt/s)/t)$ and rearranging yields the result:

$$p \ge p(1 - p) \ge \frac{1}{4} e^{-\varepsilon^2 k^2} = e^{-\Theta\left(\frac{\varepsilon}{t}\log\left(\frac{xt}{s}\right)\right)^2}.$$

$\square$

We also provide a lower bound on the variance of the distributions providing concentrated differential privacy with smooth sensitivity.

**Proposition 36.** *Let $s, t, \varepsilon > 0$. Let $Z$ be a real random variable satisfying $\mathbb{E}[Z] = 0$ and, for all $\alpha \in (1, \infty)$,*

$$\mathrm{D}_\alpha\left(e^t Z + s \,\|\, Z\right) \leq \frac{1}{2}\varepsilon^2\alpha \quad \text{and} \quad \mathrm{D}_\alpha\left(Z \,\|\, e^{-t}Z - s\right) \leq \frac{1}{2}\varepsilon^2\alpha.$$

*Then, for all $k, \ell \in \mathbb{Z}$ with $k, \ell \geq 0$ and $k + \ell > 0$,*

$$\mathbb{E}\left[Z^2\right] \geq \frac{s^2}{(1 - e^{-t})^2}\left(\frac{\left(e^{(k+\ell-1)t} - e^{(\ell-1)t} + e^{\ell t} - 1\right)^2}{e^{\varepsilon^2(k+\ell)^2} - 1} - \left(e^{\ell t} - 1\right)^2\right).$$

Setting $\ell = 0$ yields

$$\mathbb{E}\left[Z^2\right] \geq \frac{s^2}{(e^t - 1)^2} \cdot \frac{\left(e^{kt} - 1\right)^2}{e^{\varepsilon^2 k^2} - 1}.$$

*Proof.* By group privacy (Lemma 11), for all $k, \ell \in \mathbb{Z}$ with $k, \ell \geq 0$ and all $\alpha \in (1, \infty)$,

$$\mathrm{D}_\alpha\left(e^{kt}Z + s\frac{e^{kt} - 1}{e^t - 1} \,\middle\|\, e^{-\ell t}Z - s\frac{1 - e^{-\ell t}}{1 - e^{-t}}\right) \leq \frac{1}{2}\varepsilon^2(k + \ell)^2\alpha.$$

We next use a Rényi version of Pinsker's inequality [BS16, Lem. C.2]:

$$\forall X \; \forall Y \quad |\mathbb{E}[X] - \mathbb{E}[Y]| \leq \sqrt{\mathbb{E}[Y^2]\left(e^{\mathrm{D}_2(X\|Y)} - 1\right)}.$$

Setting $X = e^{kt}Z + s\frac{e^{kt}-1}{e^t-1}$ and $Y = e^{-\ell t}Z - s\frac{1-e^{-\ell t}}{1-e^{-t}}$ in the above, we have

$$s\left(\frac{e^{kt} - 1}{e^t - 1} + \frac{1 - e^{-\ell t}}{1 - e^{-t}}\right) \leq \sqrt{\mathbb{E}\left[\left(e^{-\ell t}Z - s\frac{1 - e^{-\ell t}}{1 - e^{-t}}\right)^2\right]\left(e^{\varepsilon^2(k+\ell)^2} - 1\right)}.$$

This rearranges to

$$e^{-2\ell t}\mathbb{E}\left[Z^2\right] + s^2\left(\frac{1 - e^{-\ell t}}{1 - e^{-t}}\right)^2 \geq s^2\left(\frac{e^{kt} - 1}{e^t - 1} + \frac{1 - e^{-\ell t}}{1 - e^{-t}}\right)^2\frac{1}{e^{\varepsilon^2(k+\ell)^2} - 1}$$

and

$$\mathbb{E}\left[Z^2\right] \geq s^2 e^{2\ell t}\left(\left(\frac{e^{kt} - 1}{e^t - 1} + \frac{1 - e^{-\ell t}}{1 - e^{-t}}\right)^2\frac{1}{e^{\varepsilon^2(k+\ell)^2} - 1} - \left(\frac{1 - e^{-\ell t}}{1 - e^{-t}}\right)^2\right)$$

$$= \frac{s^2}{(1 - e^{-t})^2}\left(\frac{\left(e^{(k+\ell-1)t} - e^{(\ell-1)t} + e^{\ell t} - 1\right)^2}{e^{\varepsilon^2(k+\ell)^2} - 1} - \left(e^{\ell t} - 1\right)^2\right).$$

$\square$

Figure 1: Variance of the Trimmed Mean for Various distributions as the trimming fraction is varied. The plot depicts experimental values for $n = 1001$ averaged over $10^6$ repetitions.

# 5    Trimmed Mean

For the problem of mean estimation, we use the trimmed mean as our estimator.

**Definition 37** (Trimmed Mean)**.** *For* $n, m \in \mathbb{Z}$ *with* $n > 2m \geq 0$, *define* $\mathsf{trim}_m : \mathbb{R}^n \to \mathbb{R}$ *by*

$$\mathsf{trim}_m(x) = \frac{x_{(m+1)} + x_{(m+2)} + \cdots + x_{(n-m)}}{n - 2m},$$

*where* $x_{(1)} \leq x_{(2)} \leq \cdots \leq x_{(n)}$ *denote the order statistics of* $x$.

Intuitively, the trimmed mean interpolates between the mean ($m = 0$) and the median ($m = \frac{n-1}{2}$).

## 5.1    Error of the Trimmed Mean

Before we consider privatising the trimmed mean, we look at the error introduced by the trimming itself. We focus on mean squared error relative to the mean. That is,

$$\mathop{\mathbb{E}}_{X \leftarrow \mathcal{D}^n} \left[ (\mathsf{trim}_m(X) - \mu)^2 \right],$$

where $\mu = \mathop{\mathbb{E}}_{X \leftarrow \mathcal{D}} [X]$ is the mean of the distribution $\mathcal{D}$.

We remark that mean squared error may not be the most relevant error metric for many applications. For example, the length of a confidence interval may be more relevant [KV18].

Similarly, the mean may not be the most relevant parameter to estimate. We pick this error metric as it is simple, widely-applicable, and does not require picking additional parameters (such as the confidence level).

The error of the trimmed mean depends on both the trimming fraction and also the data distribution. Figure 1 illustrates this. For Gaussian data, the optimal estimate is the empirical mean, corresponding to trimming $m = 0$ elements. This has mean squared error $\frac{1}{n}$ for $n$ samples. As the trimming fraction is increased, the error does too. At the extreme, the median of Gaussian data has asymptotic variance $\frac{\pi}{2n} \approx \frac{1.57}{n}$. However, if the data has slightly heavier tails than Gaussian data, such as Laplacian data, then trimming actually reduces variance. The Laplacian Mean has variance $\frac{2}{n}$, while the median has asymptotic variance $\frac{1}{n}$. In between these two cases is a mixture of two Gaussians with the same mean and differring variances. Here a small amount of trimming reduces the error, but a large amount of trimming increases it again, and there is an optimal trimming fraction in between.

For our main theorems we use the following two analytic bounds. The first is a strong bound for symmetric distributions, while the second is a weaker bound for asymmetric distributions.

**Proposition 38.** *Let $\mathcal{D}$ be a symmetric distribution on $\mathbb{R}$. Let $n, m \in \mathbb{Z}$ satisfy $n > 2m \geq 0$. Then $\mathsf{trim}_m(X)$ is also symmetric for $X \leftarrow \mathcal{D}^n$. Moreover,*

$$\mathop{\mathbb{E}}_{X \leftarrow \mathcal{D}^n} \left[ (\mathsf{trim}_m(X))^2 \right] \leq \frac{n}{(n - 2m)^2} \cdot \mathop{\mathbb{E}}_{X \leftarrow \mathcal{D}} \left[ X^2 \right].$$

This lemma follows from the symmetry of both the trimming and the distribution.

**Proposition 39.** *Let $n, m \in \mathbb{Z}$ satisfy $n > 2m \geq 0$. Let $X_1, \cdots, X_n$ be i.i.d. samples from a distribution with mean $\mu$ and variance $\sigma^2$. Then*

$$\mathbb{E}\left[ (\mathsf{trim}_m(X) - \mu)^2 \right] \leq \frac{n(1 + \sqrt{8m})}{(n - 2m)^2} \sigma^2 = O\left( \frac{m}{n} \sigma^2 \right).$$

We first remark about the tightness of this bound: Consider the Bernoulli distribution with mean $\mu = m/2n$. With high probability $(1 - 2^{-\Omega(m)})$, the trimming removes all the 1s. Thus $\mathbb{E}\left[ (\mathsf{trim}_m(X) - \mu)^2 \right] \geq 0.8\mu^2 = \frac{m^2}{5n^2}$. And we have $\frac{m}{n}\sigma^2 = \frac{m^2}{2n^2}(1 - \mu) \leq \frac{m^2}{2n^2}$. Thus the bound is tight up to constant factors (in the regime where $n - 2m = \Omega(n)$).

*Proof.* By affine scaling, we may assume that $\mu = 0$ and $\sigma^2 = 1$. Let $X_1, \cdots, X_n$ be i.i.d. samples from some distribution with mean zero and variance one. Let $\sigma$ be a permutation on $[n]$ such that $X_{\sigma(1)} \leq X_{\sigma(2)} \leq \cdots \leq X_{\sigma(n)}$ and $\sigma$ is uniform (that is, ties are broken randomly).

We make $\sigma$ explicit in this proof so that we can reason about its uniform distribution and we can talk about $\sigma^{-1}(i)$, which is the index of $X_i$ in the sorted order.

For $s \subset [n]$, let $\sigma_s$ denote the restriction of $\sigma$ to $s$. That is, $s = \{\sigma_s(1), \sigma_s(2), \cdots, \sigma_s(|s|)\}$ and $\sigma_s^{-1}(i) \leq \sigma_s^{-1}(j) \iff \sigma^{-1}(i) \leq \sigma^{-1}(j)$ for all $i, j \in s$.

Let

$$Y = (n - 2m)\mathsf{trim}_m(X) = \sum_{i=m+1}^{n-m} X_{\sigma(i)} = \sum_{i=1}^{n} (1 - A_i)X_i,$$

where $A_i \in \{0, 1\}$ is defined by

$$1 - A_i = 1 \iff m + 1 \le \sigma^{-1}(i) \le n - m.$$

Note that $\mathbb{E}[A_i] = \frac{2m}{n}$ for all $i \in [n]$.

Our goal is to bound

$$
\begin{aligned}
\mathbb{E}\left[Y^2\right] &= \sum_{i,j \in [n]} \mathbb{E}\left[(1 - A_i)X_i(1 - A_j)X_j\right] \\
&= n - \sum_{i,j \in [n]} \mathbb{E}\left[X_i X_j (A_i + A_j - A_i A_j)\right] \\
&= n - \sum_{i \in [n]} \mathbb{E}\left[X_i^2 A_i\right] \\
&\quad - \sum_{i,j \in [n]: i \neq j} \mathbb{E}\left[X_i X_j \tilde{A}_i^j\right] + \mathbb{E}\left[X_i X_j \tilde{A}_j^i\right] + \mathbb{E}\left[X_i X_j (A_i - \tilde{A}_i^j + A_j - \tilde{A}_j^i - A_i A_j)\right],
\end{aligned}
$$

where $\tilde{A}_i^j$ will be defined next. For $i, j \in [n]$ with $i \neq j$,

$$1 - \tilde{A}_i^j = 1 \iff m + 1 \le \sigma^{-1}_{[n] \setminus \{j\}}(i) \le (n - 1) - m.$$

Since $\sigma^{-1}(i) - 1 \le \sigma^{-1}_{[n] \setminus \{j\}}(i) \le \sigma^{-1}(i)$, we have $1 - \tilde{A}_i^j \le 1 - A_i$ for all $i, j \in [n]$ with $i \neq j$.

We also have $\sigma^{-1}_{[n] \setminus \{j\}}(i) = \sigma^{-1}(i) \iff \sigma^{-1}(j) > \sigma^{-1}(i)$ for all $i, j \in [n]$ with $i \neq j$. Now, for $i, j \in [n]$ with $i \neq j$, we have

$$
\begin{aligned}
\tilde{A}_i^j \neq A_i &\iff \sigma^{-1}_{[n] \setminus \{j\}}(i) = m = \sigma^{-1}(i) - 1 \ \ \text{OR} \ \ \sigma^{-1}_{[n] \setminus \{j\}}(i) = n - m = \sigma^{-1}(i) \\
&\iff \left(\sigma^{-1}(i) = m + 1 \ \text{AND} \ \sigma^{-1}(j) < \sigma^{-1}(i)\right) \ \ \text{OR} \ \ \left(\sigma^{-1}(i) = n - m \ \text{AND} \ \sigma^{-1}(j) > \sigma^{-1}(i)\right) \\
&\iff \left(\sigma^{-1}(i) = m + 1 \ \text{AND} \ \sigma^{-1}(j) < m + 1\right) \ \ \text{OR} \ \ \left(\sigma^{-1}(i) = n - m \ \text{AND} \ \sigma^{-1}(j) > n - m\right).
\end{aligned}
$$

Since $\sigma$ is a uniformly random permutation, we conclude that, for $i, j \in [n]$ with $i \neq j$,

$$\mathbb{P}\left[\tilde{A}_i^j \neq A_i\right] \le \mathbb{P}\left[\sigma^{-1}(j) < \sigma^{-1}(i) = m + 1\right] + \mathbb{P}\left[\sigma^{-1}(j) > \sigma^{-1}(i) = n - m\right] \le \frac{2m}{n(n-1)}.$$

Now we observe that (by construction) the pair $(X_i, \tilde{A}_i^j)$ is independent from $X_j$ for all $i, j \in [n]$ with $i \neq j$.

We return to our calculation:

$$\mathbb{E}\left[Y^2\right] \leq n - \sum_{i \in [n]} \mathbb{E}\left[X_i^2 0\right]$$

$$- \sum_{i,j \in [n]: i \neq j} \mathbb{E}\left[X_i \tilde{A}_i^j\right] \mathbb{E}\left[X_j\right] + \mathbb{E}\left[X_i\right] \mathbb{E}\left[X_j \tilde{A}_j^i\right] + \mathbb{E}\left[X_i X_j (A_i - \tilde{A}_i^j + A_j - \tilde{A}_j^i - A_i A_j)\right]$$

$$= n - \sum_{i,j \in [n]: i \neq j} \mathbb{E}\left[X_i X_j (A_i - \tilde{A}_i^j + A_j - \tilde{A}_j^i - A_i A_j)\right]$$

$$\leq n + \sum_{i,j \in [n]: i \neq j} \sqrt{\mathbb{E}\left[(X_i X_j)^2\right] \mathbb{E}\left[(A_i - \tilde{A}_i^j + A_j - \tilde{A}_j^i - A_i A_j)^2\right]}$$

$$\leq n + \sum_{i,j \in [n]: i \neq j} \sqrt{2\mathbb{E}\left[\tilde{A}_i^j - A_i + \tilde{A}_j^i - A_j + A_i A_j\right]}$$

$$\leq n + n(n-1)\sqrt{2\left(\frac{2m}{n(n-1)} + \frac{2m}{n(n-1)} + \frac{2m}{n} \cdot \frac{2m-1}{n-1}\right)}$$

$$\leq n + \sqrt{8}nm.$$

Finally,

$$\mathbb{E}\left[(\mathsf{trim}_m(X))^2\right] = \frac{1}{(n-2m)^2}\mathbb{E}\left[Y^2\right] \leq \frac{n(1+\sqrt{8}m)}{(n-2m)^2}$$

$\square$

## 5.2 Sensitivity of Trimmed Mean

The other key property we need is that the trimmed mean has low local and smooth sensitivity.

**Proposition 40.** *Let $a, b, t \in \mathbb{R}$ with $a < b$ and $t \geq 0$ and $n, m, k \in \mathbb{Z}$ with $n > 2m \geq 0$ and $k \geq 0$ and $x \in \mathbb{R}^n$. Denote $x_1, x_2, \cdots, x_n$ in sorted order as $x_{(1)} \leq x_{(2)} \leq \cdots \leq x_{(n)}$. The local sensitivity of the trimmed mean at $x$ is*

$$\mathsf{LS}_{\mathsf{trim}_m}(x) = \frac{\max\{x_{(n-m+1)} - x_{(m+1)}, x_{(n-m)} - x_{(m)}\}}{n - 2m}$$

*and at distance $k$ it is*

$$\mathsf{LS}_{\mathsf{trim}_m}^x(k) = \begin{cases} \frac{\max_{\ell=0}^{k+1} x_{(n-m+1+k-\ell)} - x_{(m+1-\ell)}}{n-2m} & \text{if } k < m \\ \infty & \text{if } k \geq m \end{cases}.$$

*The $t$-smooth sensitivity of the trimmed mean restricted to inputs in $[a, b]$ – that is, $\mathsf{trim}_m : [a, b]^n \to [a, b]$ – is*

$$\mathsf{S}_{\mathsf{trim}_m}^t(x) = \frac{1}{n - 2m} \max_{k=0}^{n} e^{-kt} \max_{\ell=0}^{k+1} x_{(n-m+1+k-\ell)} - x_{(m+1-\ell)},$$

*where we define $x_{(i)} = a$ for $i \leq 0$ and $x_{(i)} = b$ for $i > n$.*

The proof of this is a direct extension of the analysis of the smooth sensitivity of the median by Nissim, Raskhodnikova, and Smith [NRS07]. There is a $O(n \log n)$-time algorithm for computing the smooth sensitivity.

# 6   Average-Case Mean Estimation via Smooth Sensitivity of Trimmed Mean

Having compiled the relevant tools in the previous sections, we turn to applying them to the problem of mean estimation. We consider an average-case distributional setting. We have an unknown distribution $\mathcal{D}$ on $\mathbb{R}$ and our goal is to estimate the mean $\mu = \mathop{\mathbb{E}}_{X \leftarrow \mathcal{D}}[X]$, given $n$ independent samples $X_1, \cdots, X_n$ from $\mathcal{D}$.

Our non-private comparison point is the (un-trimmed) empirical mean $\overline{X} = \frac{1}{n} \sum_{i=1}^{n} X_i$. This is unbiased – that is, $\mathop{\mathbb{E}}_{X \leftarrow \mathcal{D}^n}[\overline{X}] = \mu$ and has variance $\mathop{\mathbb{E}}_{X \leftarrow \mathcal{D}^n}[(\overline{X} - \mu)^2] = \frac{\sigma^2}{n}$, where $\sigma^2 = \mathop{\mathbb{E}}_{X \leftarrow \mathcal{D}}[(X - \mu)^2]$.

We make the assumption that some loose bound $\mu \in [a, b]$ is known. Our results will only pay logarithmically in $b - a$, so this bound need not be tight. In general, some dependence on this range is required.

In our situation the inputs may be unbounded. This means the trimmed mean has infinite global sensitivity and thus infinite smooth sensitivity. Thus we must do something to control the sensitivity. We apply truncation:

**Definition 41** (Truncation). *For $a, b, x \in \mathbb{R}$ with $a < b$, define*

$$[x]_{[a,b]} = \begin{cases} b & \text{if } x > b \\ x & \text{if } a \leq x \leq b \\ a & \text{if } x < a \end{cases} .$$

*For $x \in \mathbb{R}^n$ and $a < b$, define $[x]_{[a,b]} = ([x_1]_{[a,b]}, [x_2]_{[a,b]}, \cdots, [x_n]_{[a,b]})$.*

## 6.1   Truncation of Inputs

By truncating inputs before applying the trimmed mean, we obtain the following error bound. This holds for symmetric and subgaussian distributions.

The key is that, if we know that the distribution $\mathcal{D}$ is $\sigma$-subgaussian and it's mean lies within $[a, b]$, then we can truncating the inputs to the range $[a - O(\sigma \log n), b + O(\sigma \log n)]$ without significantly affecting them.

**Proposition 42.** *Let $\mathcal{D}$ be a symmetric $\overline{\sigma}$-subgaussian distribution on $\mathbb{R}$. Let $\mu = \mathop{\mathbb{E}}_{X \leftarrow \mathcal{D}}[X]$*

and $\sigma^2 = \mathop{\mathbb{E}}_{X \leftarrow \mathcal{D}}[(X - \mu)^2]$. Let $a < \mu < b$. t $n, m \in \mathbb{Z}$ satisfy $n > 2m \geq 0$. Then

$$\mathop{\mathbb{E}}_{X \leftarrow \mathcal{D}^n}\left[\left(\mathsf{trim}_m\left([X]_{[a,b]}\right) - \mu\right)^2\right] \leq \left(\sqrt{\frac{n\sigma^2}{(n-2m)^2}} + \sqrt{\frac{2n\overline{\sigma}^2}{n-2m}\left(e^{-(b-\mu)^2/2\overline{\sigma}^2} + e^{-(\mu-a)^2/2\overline{\sigma}^2}\right)}\right)^2$$

$$= \frac{\sigma^2}{n}\left(1 + \frac{2m}{n-2m} + \sqrt{\frac{\overline{\sigma}^2}{\sigma^2}\frac{2n^2}{n-2m}(e^{-(b-\mu)^2/2\overline{\sigma}^2} + e^{-(\mu-a)^2/2\overline{\sigma}^2})}\right)^2$$

$$= \frac{\sigma^2}{n}\left(1 + O\left(\frac{m}{n}\right)\right),$$

where the final asymptotic statement assumes $n - 2m = \Omega(n)$ and $\min\{b - \mu, \mu - a\} \geq O(\overline{\sigma}\log n)$ and $\overline{\sigma}^2 = O(\sigma^2)$.

We remark that if $\mathcal{D}$ is not subgaussian, but rather subexponential then a similar bound can be proved. This result is simply meant to be indicative of what is possible.

*Proof.* We may assume $\mu = 0$ without loss of generality. The result follows by combining Proposition 38 and Lemma 43 with the following reformulation of the Cauchy-Schwartz inequality.

$$\forall X, Y \quad \mathbb{E}\left[(X + Y)^2\right] \leq \left(\sqrt{\mathbb{E}[X^2]} + \sqrt{\mathbb{E}[Y^2]}\right)^2.$$

$\square$

**Lemma 43.** *Let $\mathcal{D}$ be a centered $\sigma$-subgaussian distribution on $\mathbb{R}$. Let $n, m \in \mathbb{Z}$ satisfy $n > 2m \geq 0$. Let $a < 0 < b$. Then*

$$\mathop{\mathbb{E}}_{X \leftarrow \mathcal{D}^n}\left[\left(\mathsf{trim}_m\left([X]_{[a,b]}\right) - \mathsf{trim}_m(X)\right)^2\right] \leq \frac{2n}{n-2m} \cdot \sigma^2 \cdot \left(e^{-b^2/2\sigma^2} + e^{-a^2/2\sigma^2}\right)$$

*Proof.* We begin with the standard tail bound of subgaussians: For $t, x \geq 0$,

$$\mathop{\mathbb{P}}_{X \leftarrow \mathcal{D}}[X \geq x] = \mathop{\mathbb{E}}_{X \leftarrow \mathcal{D}}[\mathbb{I}[X \geq x]] \leq \mathop{\mathbb{E}}_{X \leftarrow \mathcal{D}}\left[e^{t(X-x)}\right] \leq e^{\sigma^2 t^2/2 - tx} = e^{-x^2/2\sigma^2},$$

where the final inequality follows by setting $t = x/\sigma^2$ to minimize the expression. Similarly

$\mathop{\mathbb{P}}\limits_{X \leftarrow \mathcal{D}}[X \leq x] \leq e^{-x^2/2\sigma^2}$ for $x \leq 0$. Next we apply this to the quantitity at hand:

$$\mathop{\mathbb{E}}_{X \leftarrow \mathcal{D}^n} \left[ \left( \text{trim}_m \left( [X]_{[a,b]} \right) - \text{trim}_m(X) \right)^2 \right] \leq \frac{1}{n-2m} \sum_{\ell=m+1}^{n-m} \mathop{\mathbb{E}}_{X \leftarrow \mathcal{D}^n} \left[ \left( \left[ X_{(\ell)} \right]_{[a,b]} - X_{(\ell)} \right)^2 \right]$$

$$\leq \frac{1}{n-2m} \sum_{\ell=1}^{n} \mathop{\mathbb{E}}_{X \leftarrow \mathcal{D}^n} \left[ \left( [X_\ell]_{[a,b]} - X_\ell \right)^2 \right]$$

$$(f_\mathcal{D} \text{ is the density of } \mathcal{D}) \quad = \frac{n}{n-2m} \left( \int_b^\infty (x-b)^2 f_\mathcal{D}(x) \mathrm{d}x + \int_{-\infty}^a (a-x)^2 f_\mathcal{D}(x) \mathrm{d}x \right)$$

$$(\text{integration by parts}) \quad = \frac{n}{n-2m} \left( \begin{array}{c} \int_b^\infty 2(x-b) \mathop{\mathbb{P}}\limits_{X \leftarrow \mathcal{D}}[X \geq x] \mathrm{d}x \\ + \int_{-\infty}^a 2(a-x) \mathop{\mathbb{P}}\limits_{X \leftarrow \mathcal{D}}[X \leq x] \mathrm{d}x \end{array} \right)$$

$$\leq \frac{2n}{n-2m} \left( \begin{array}{c} \int_b^\infty (x-b) e^{-x^2/2\sigma^2} \mathrm{d}x \\ + \int_{-\infty}^a (a-x) e^{-x^2/2\sigma^2} \mathrm{d}x \end{array} \right)$$

$$= \frac{2n}{n-2m} \int_0^\infty x \left( e^{-(x+b)^2/2\sigma^2} + e^{-(x-a)^2/2\sigma^2} \right) \mathrm{d}x$$

$$\leq \frac{2n}{n-2m} \int_0^\infty x \cdot e^{-x^2/2\sigma^2} \cdot \left( e^{-b^2/2\sigma^2} + e^{-a^2/2\sigma^2} \right) \mathrm{d}x$$

$$= \frac{2n}{n-2m} \cdot \sigma^2 \cdot \left( e^{-b^2/2\sigma^2} + e^{-a^2/2\sigma^2} \right).$$

$\square$

Next we turn to analyzing the smooth sensitivity of the trimmed mean with truncated inputs.

**Lemma 44.** *Let $\mathcal{D}$ be a $\sigma$-subgaussian distribution on $\mathbb{R}$. Let $a < 0 < b$. Then*

$$\mathop{\mathbb{E}}_{X \leftarrow \mathcal{D}^n} \left[ \left( S^t_{\text{trim}_m \left( [\cdot]_{[a,b]} \right)}(X) \right)^2 \right] \leq \frac{8\sigma^2 \log n + e^{-2mt}(b-a)^2}{(n-2m)^2}.$$

*Proof.* By Proposition 40,

$$S^t_{\text{trim}_m \left( [\cdot]_{[a,b]} \right)}(x) = \frac{1}{n-2m} \max_{k=0}^{n} e^{-kt} \max_{\ell=0}^{k+1} x_{(n-m+1+k-\ell)} - x_{(m+1-\ell)} \leq \frac{\max\{x_{(n)} - x_{(1)}, e^{-mt} \cdot (b-a)\}}{n-2m},$$

where the inequality follows from the fact that $x_{(n-m+1+k-\ell)} - x_{(m+1-\ell)} \leq x_{(n)} - x_{(1)}$ when $k < m$ and $x_{(n-m+1+k-\ell)} - x_{(m+1-\ell)} \leq b - a$ when $k \geq m$. Thus

$$\mathop{\mathbb{E}}_{X \leftarrow \mathcal{D}^n} \left[ \left( S^t_{\text{trim}_m \left( [\cdot]_{[a,b]} \right)}(X) \right)^2 \right] \leq \frac{1}{(n-2m)^2} \mathop{\mathbb{E}}_{X \leftarrow \mathcal{D}^n} \left[ (X_{(n)} - X_{(1)})^2 + e^{-2mt}(b-a)^2 \right]$$

$$\leq \frac{8\sigma^2 \log(2n) + e^{-2mt}(b-a)^2}{(n-2m)^2},$$

where the final inequality follows from Lemma 45 and the fact that $(x-y)^2 \leq 4 \max\{x^2, y^2\}$ for all $x, y \in \mathbb{R}$. $\square$

**Lemma 45** ([FS18, Lem. 4.5])**.** *Let $\mathcal{D}$ be a $\sigma$-subgaussian distribution. Then $\underset{X \leftarrow \mathcal{D}^n}{\mathbb{E}} \left[\max_{i=1}^n X_i^2\right] \leq 2\sigma^2 \log(2n)$.*

Combining Proposition 42 and Lemma 44 yields the following result.

**Theorem 46.** *Let $s, t, \varepsilon > 0$. Let $Z$ be a centered distribution with the property that*

$$\left. \begin{array}{l} \mathrm{D}_\alpha \left(Z \| e^{\pm t} Z \pm s\right) \\ \mathrm{D}_\alpha \left(e^{\pm t} Z \pm s \| Z\right) \end{array} \right\} \leq \frac{1}{2}\alpha\varepsilon^2$$

*for all $\alpha \in (1, \infty)$.*
   *Let $n, m \in \mathbb{Z}$ with $n > 2m \geq 0$. Let $a < b$ and $c, \sigma, \overline{\sigma} > 0$.*
   *Define a randomized algorithm $M : \mathbb{R}^n \to \mathbb{R}$ by*

$$M(x) = \mathsf{trim}_m\left([x]_{[a,b]}\right) + \frac{1}{s}\mathsf{S}^t_{\mathsf{trim}_m\left([\cdot]_{[a,b]}\right)}(X) \cdot Z.$$

*Then $M$ is $\frac{1}{2}\varepsilon^2$-CDP and has the following property. Let $\mathcal{D}$ be a distribution that is symmetric about its mean $\mu \in [a + \overline{\sigma}c, b - \overline{\sigma}c]$ and has variance $\sigma^2$ and is $\overline{\sigma}$-subgaussian. Then*

$$\underset{X \leftarrow \mathcal{D}^n}{\mathbb{E}}\left[(M(X) - \mu)^2\right] \leq \frac{\sigma^2}{n}\left(1 + \frac{2m}{n - 2m} + \sqrt{\frac{\overline{\sigma}^2}{\sigma^2}\frac{2n^2}{n - 2m}\left(e^{-(b-\mu)^2/2\overline{\sigma}^2} + e^{-(\mu-a)^2/2\overline{\sigma}^2}\right)}\right)^2$$

$$+ \frac{8\overline{\sigma}^2\log(2n) + e^{-2mt}(b - a)^2}{(n - 2m)^2 s^2} \cdot \mathsf{Var}\left[Z\right].$$

Combining Theorem 46 with the distributions from Section 3 yields the following results. The first is a simpler case when the variance is known and the second is for when it is unknown.

**Corollary 47.** *Let $\varepsilon, \sigma > 0$, $a < b$, and $n \in \mathbb{Z}$ with $n \geq O(\log((b - a)/\sigma)/\varepsilon)$. There exists a $\frac{1}{2}\varepsilon^2$-CDP algorithm $M : \mathbb{R}^n \to \mathbb{R}$ such that the following holds. Let $\mathcal{D}$ be a $\sigma$-subgaussian distribution that is symmetric about its mean $\mu \in [a, b]$. Then*

$$\underset{X \leftarrow \mathcal{D}^n}{\mathbb{E}}\left[(M(X) - \mu)^2\right] \leq \frac{\sigma^2}{n} + \frac{\sigma^2}{n^2} \cdot O\left(\frac{\log((b - a)/\sigma)}{\varepsilon} + \frac{\log n}{\varepsilon^2}\right).$$

**Corollary 48.** *Let $\varepsilon \in (0, 1)$, $0 < \underline{\sigma} \leq \overline{\sigma}$, $a < b$, and $n \in \mathbb{Z}$ with $n \geq O(\log((b-a)\overline{\sigma}/\underline{\sigma}^2)/\varepsilon)$. There exists a $\frac{1}{2}\varepsilon^2$-CDP algorithm $M : \mathbb{R}^n \to \mathbb{R}$ such that the following holds. Let $\mathcal{D}$ be a distrbution that is symmetric about its mean $\mu \in [a, b]$ and is $\sigma$-subgaussian with $\underline{\sigma} \leq \sigma \leq \overline{\sigma}$. Then*

$$\underset{X \leftarrow \mathcal{D}^n}{\mathbb{E}}\left[(M(X) - \mu)^2\right] \leq \frac{\sigma^2}{n} + \frac{\sigma^2}{n^2} \cdot O\left(\frac{\log((b - a)/\underline{\sigma}) + \log(\overline{\sigma}/\underline{\sigma})}{\varepsilon} + \frac{\log n}{\varepsilon^2}\right).$$

## 6.2 Truncation of Outputs

Rather than truncating the inputs to the trimmed mean, we can truncate the output. This is useful for heavier-tailed distributions and is also simpler to analyze. Indeed, the truncation only reduces error:

**Lemma 49.** *Let $a \le \mu \le b$ and let $X$ be a random variable. Then*

$$\mathbb{E}\left[[X - \mu]_{[a,b]}^2\right] \le \mathbb{E}\left[(X - \mu)^2\right].$$

Truncation of the output also controls the smooth sensitivity:

**Lemma 50.** *Let $n, m \in \mathbb{Z}$ with $n > 2m \ge 0$. Let $t > 0$ and $a < b$. For $x \in \mathbb{R}^n$, we have*

$$\mathsf{S}^t_{[\mathrm{trim}_m(\cdot)]_{[a,b]}}(x) \le \max\left\{\frac{x_{(n)} - x_{(1)}}{n - 2m}, e^{-mt}(b - a)\right\}.$$

*Let $\mathcal{D}$ be a distribution with variance $\sigma^2$. Then*

$$\mathop{\mathbb{E}}_{X \leftarrow \mathcal{D}^n}\left[\left(\mathsf{S}^t_{[\mathrm{trim}_m(\cdot)]_{[a,b]}}(X)\right)^2\right] \le \sigma^2 \frac{2n}{(n - 2m)^2} + e^{-2mt}(b - a)^2.$$

This should be contrasted with Lemma 44. The proof is also nearly identical.

*Proof.* Proposition 40 and the fact that $\mathsf{LS}^k_{[\mathrm{trim}_m(\cdot)]_{[a,b]}}(x) \le \min\{\mathsf{LS}^k_{\mathrm{trim}_m}(x), b - a\}$ give

$$\mathsf{LS}^x_{[\mathrm{trim}_m(\cdot)]_{[a,b]}}(k) \le \begin{cases} \frac{\max_{\ell=0}^{k+1} x_{(n-m+1+k-\ell)} - x_{(m+1-\ell)}}{n-2m} & \text{if } k < m \\ b - a & \text{if } k \ge m \end{cases}.$$

We thus obtain the first part of the result from Definition 14 and the fact that $x_{(n-m+1+k-\ell)} - x_{(m+1-\ell)} \le x_{(n)} - x_{(1)}$ for $0 \le \ell \le k + 1 \le m$, namely

$$\mathsf{S}^x_f(t) = \max_{k \ge 0} e^{-tk} \mathsf{LS}^k_f(x) \le \max\left\{\frac{x_{(n)} - x_{(1)}}{n - 2m}, e^{-mt}(b - a)\right\}.$$

Next we use the inequality $(x_{(n)} - x_{(1)})^2 \le 2 \sum_{i=1}^n x_i^2$ to obtain

$$\mathop{\mathbb{E}}_{X \leftarrow \mathcal{D}^n}\left[\left(\mathsf{S}^t_{[\mathrm{trim}_m(\cdot)]_{[a,b]}}(X)\right)^2\right] \le \frac{\mathbb{E}\left[(X_{(n)} - X_{(1)})^2\right]}{(n - 2m)^2} + e^{-2mt}(b-a)^2 \le \frac{2n\sigma^2}{(n - 2m)^2} + e^{-2mt}(b-a)^2.$$

$\square$

Combining Proposition 39, Lemma 49, and Lemma 50 yields the following result.

**Theorem 51.** *Let $s, t, \varepsilon > 0$. Let $Z$ be a centered distribution with the property that*

$$\left.\begin{array}{l} \mathrm{D}_\alpha\left(Z \| e^{\pm t} Z \pm s\right) \\ \mathrm{D}_\alpha\left(e^{\pm t} Z \pm s \| Z\right) \end{array}\right\} \leq \frac{1}{2} \alpha \varepsilon^2$$

*for all $\alpha \in (1, \infty)$.*

*Let $n, m \in \mathbb{Z}$ with $n > 2m \geq 0$. Let $a < b$ and $\sigma > 0$. Define a randomized algorithm $M : \mathbb{R}^n \to \mathbb{R}$ by*

$$M(x) = \left[\mathsf{trim}_m(x)\right]_{[a,b]} + \frac{1}{s} \mathsf{S}^t_{[\mathsf{trim}_m(\cdot)]_{[a,b]}}(x) \cdot Z.$$

*Then $M$ is $\frac{1}{2}\varepsilon^2$-CDP and has the following property. Let $\mathcal{D}$ be a distribution with mean $\mu \in [a, b]$ and variance $\sigma^2$. Then*

$$\mathop{\mathbb{E}}_{X \leftarrow \mathcal{D}^n}\left[(M(X) - \mu)^2\right] \leq \frac{n(1 + \sqrt{8m})}{(n - 2m)^2}\sigma^2 + \left(\sigma^2 \frac{2n}{(n - 2m)^2} + e^{-2mt}(b - a)^2\right) \cdot \frac{1}{s^2} \cdot \mathsf{Var}\,[Z]$$

$$\leq \frac{\sigma^2}{n} \cdot O\left(m + \frac{\mathsf{Var}\,[Z]}{s^2}\right) + e^{-2mt}(b - a)^2.$$

Combining Theorem 51 with the distributions from Section 3 yields the following result.

**Corollary 52.** *Let $\varepsilon > 0$ and $0 < \underline{\sigma}$ and $a < b$. Let $n \geq O(\log(n(b - a)/\underline{\sigma})/\varepsilon)$. Then there exists a $\frac{1}{2}\varepsilon^2$-CDP algorithm $M : \mathbb{R}^n \to \mathbb{R}$ such that the following holds. Let $\mathcal{D}$ be a distribution with mean $\mu \in [a, b]$ and variance $\sigma^2 \geq \underline{\sigma}^2$. Then*

$$\mathop{\mathbb{E}}_{X \leftarrow \mathcal{D}^n}\left[(M(X) - \mu)^2\right] \leq \frac{\sigma^2}{n} \cdot O\left(\frac{\log(n(b - a)/\underline{\sigma})}{\varepsilon} + \frac{1}{\varepsilon^2}\right).$$

# 7 Experimental Results

We perform an experimental evaluation of our methods, specifically the combination of the trimmed mean with various smooth sensitivity distributions applied to Gaussian data. The results are shown in Figures 2, 3, 4, 5, & 6.

## 7.1 Experimental Setup

We explain the experimental setup and parameter choices below.

- **Data:** Our data is sampled from a univariate Gaussian distribution. Gaussian is a natural data distribution. However, as shown in Figure 1, the trimmed mean performs better on heavier tailed distributions. That is to say, we would expect our results to be better for non-Gaussian data.

  The variance of our data is set to $\sigma^2 = 1$ and we set the mean to $\mu = 0$. The truncation interval is set conservatively to $[a, b] = [-50, 1050]$. The data is truncated before applying the trimmed mean.

Figure 2: Excesss variance of the private trimmed mean with smooth sensitivity for $N(0,1)$ data. Here $n = 201$ and results are averaged over $10^6$ repetitions.

Figure 3: Excesss variance of the private trimmed mean with smooth sensitivity for $N(0,1)$ data. Here $n = 501$ and results are averaged over $10^6$ repetitions.

Figure 4: Excesss variance of the private trimmed mean with smooth sensitivity for $N(0,1)$ data. Here $n = 1001$ and results are averaged over $10^6$ repetitions.

Figure 5: Excesss variance of the private trimmed mean with smooth sensitivity for $N(0,1)$ data. Here $n = 5001$ and results are averaged over $10^6$ repetitions.

Figure 6: Excesss variance of the private trimmed mean with smooth sensitivity for $N(0,1)$ data. Here $n = 5001$ and results are averaged over $10^6$ repetitions. This plot has smaller privacy parameters than the others.

- **Error:** We measure the variance or mean squared error of the various algorithms. That is,

$$\sigma^2 = \mathop{\mathbb{E}}_{X \leftarrow N(\mu,1)^n} \left[ \left( \mathsf{trim}_m \left( [X]_{[a,b]} \right) + \mathsf{S}^t_{\mathsf{trim}_m \left( [\cdot]_{[a,b]} \right)}(X) \cdot Z - \mu \right)^2 \right],$$

  where $Z$ is an appropriately-scaled distribution suited for providing differential privacy when scaled to $t$-smooth sensitivity. For scaling, we multiply by $n$ and subtract 1. Subtracting 1 corresponds to the variance of the sample mean, which is the optimal non-private error. Multiplying by $n$ allows for a comparison of different values of $n$, as it normalizes by the correct convergence rate. So we plot the normalized excess variance $\sigma^2 \cdot n - 1$ (on a logarithmic scale).

- **Algorithms:** We compare our three noise distributions against three other algorithms. Three further comparison points are provided: global sensitivity with truncation, our lower bound, and the non-private error of the trimmed mean. We explain each of the lines below.

  - `LLN`: We evaluate the Laplace Log-Normal distribution from Section 3.1. The plot uses the privacy analysis given in Theorem 18.
  - `ULN`: We evaluate the Uniform Log-Normal distribution from Section 3.2. The plot uses the privacy analysis given in Theorem 22.

<center>37</center>

- **arshinhN**: We evaluate the Arsinh-Normal distribution from Section 3.3. The plot uses the privacy analysis given in Theorem 25.

- **Student's T**: We evaluate the Student's T distribution from Section 3.4. The plot uses the privacy analysis given in Theorem 31. We set the degrees of freedom parameter to $d = 3$.

- **trim non-priv**: We plot the line where zero noise is added for privacy. The only source of error is the trimmed mean itself. This comparison point is useful as it illustrates the fact that, in many cases, most of the error is not coming from the privacy-preserving noise.

- **global sens**: We plot the error that would be attained by truncating the data and then using this to bound global sensitivity and add Gaussian noise. This is the baseline algorithm which we compare to. Note that the comparison here depends significantly on the truncation interval $[a, b]$.

- **lower bound**: We plot the lower bound on variance given by Proposition 36. No smooth sensitivity-based algorithm can beat this (although a completely different approach might).

- **Lap**: We compare to Laplace noise. This was suggested in the original work of Nissim, Raskhodnikova, and Smith [NRS07]. The plot uses the privacy analysis given in Theorem 34.

- **N**: We compare to Gaussian noise. This was analyzed in prior work [BDRS18]; see Lemma 33.

We note that the Cauchy distribution is not included in this comparison because it has infinite variance.

- **Privacy:** The algorithms we compare satisfy different variants of differential privacy. As such, it is not possible to give a completely fair comparison. Our new distributions satisfy concentrated differential privacy, whereas the Student's T distribution satisfies pure differential privacy. Laplace and Gaussian noise satisfy approximate differential privacy or truncated concentrated differential privacy.

  To provide the fairest possible comparison, we pick a $\varepsilon$ value and then compare $(\varepsilon, 0)$-differential privacy with relaxations thereof. Namely, we compare $(\varepsilon, 0)$-differential privacy with $\frac{1}{2}\varepsilon^2$-CDP, $(\frac{1}{2}\varepsilon^2, 10)$-tCDP, and $(\varepsilon, 10^{-6})$-differential privacy. Each of these is implied by $(\varepsilon, 0)$-differential privacy and the implication is fairly tight in the sense that that these definitions intuitively seem to provide a roughly similar level of privacy.

  Our plots use the values $\varepsilon = 1$ or $\varepsilon = 0.2$.

- **Other parameters:** Aside from the privacy parameters ($\varepsilon$ etc.) and the dataset size ($n$), we must choose the trimming level ($m$) and the smoothing parameter ($t$). (Note that, given the privacy parameters and smoothing parameter, the scale parameter ($s$) is chosen to be as large as possible in order to minimize the noise magnitude.)

Our plots show a range of trimming levels on the horizontal axis. We numerically optimized the smoothing parameter. Specifically, the smooth sensitivity is evaluated for 150 values of the smoothing parameter (ranging from $t = 9$ to $t = 10^{-9}$, roughly evenly spaced on a logarithmic scale) and whichever attains the lowest variance for the given algorithm and other parameter values is used.

Finally, several of our distributions have a shape parameter, which we set as follows. For Laplace Log-Normal, we numerically optimize $\sigma$; see Section 3.1.1. For Uniform Log-Normal, we set $\sigma = \sqrt{2}$, which is the smallest value permitted by our analysis (Theorem 22). For Arsinh-Normal, we set $\sigma = 2/\sqrt{3}$, which minimizes one of the terms in the analytical bound (Theorem 25). For Student's T, we set the degrees of freedom to 3 (the smallest integer with finite variance).

## 7.2 Experimental Discussion & Comparison

**Overall Performance:** The experimental results demonstrate that for relatively moderate parameter settings ($n = 201$ and $\varepsilon = 1$ depicted in Figure 2) it is possible to privately estimate the mean with variance that is only a factor of two higher than non-privately. For $n = 1001$, it is possible to drive this excess variance down to 10%. Indeed, in these settings, the additional error introduced by trimming is more significant than that introduced by the privacy-preserving noise.

We remark that the data for these experiments is perfectly Gaussian. If the data deviates from this ideal, the robustness of the trimmed mean may actually be beneficial for accuracy (and not just privacy). Figure 1 shows that for some natural distributions the trimming does reduce variance.

**Comparison of Algorithms:** The results show that different algorithms perform better in different parameter regimes. However, generally, the Laplace Log-Normal distribution has the lowest variance, closely followed by the Student's T distribution. The Arsinh-Normal distribution performs adequately, but the Uniform Log-Normal distribution performs poorly. The Laplace and Gaussian distributions from prior work often perform substantially worse than our distributions, but are better or similar in many parameter settings.

Note that the different algorithms satisfy slightly different privacy guarantees and also have very different tail behaviours. Since the variance of many of the algorithms is broadly similar, the choice of which algorithm is truly best will depend on these factors.

If the stronger pure differential privacy guarantee is preferable, the Student's T distribution is likely best. However, this has no third moment and consequently heavy tails. This makes it bad if, for example, the goal is a confidence interval, rather than a point estimate of the mean. The lightest tails are provided by the Gaussian, but this only satisfies the weaker truncated CDP or approximate differential privacy definitions. Laplace Log-Normal is in between – it satisfies the strong concentrated differential privacy definition and has quasipolynomial tails and all its moments are finite.

## Footnotes

[1] Assuming an a priori bound on the the mean is necessary to guarantee CDP. However, such a bound can, under reasonable assumptions, be discovered by an $(\varepsilon, \delta)$-DP algorithm [KV18].

[2]Specifically, the differentially private test measures how far (in terms of the number of dataset elements that need to be changed) the input dataset is from any dataset with local sensitivity higher than the proposed bound. The test is only passed if the estimated distance is large enough to be confident that it is truly nonzero, despite the noise added for privacy. Thus, for the algorithm to succeed, we need the bound to hold not just for the realized dataset instance, but also all nearby datasets.