[Reviews · NeurIPS 2019]

Reviewer 1



Overall the results are somewhat interesting. Smoothed sensitivity is a standard technique for achieving differential privacy; although most interesting smoothed sensitivity algorithms tend to be computationally hard, still the results would be interesting I believe. The paper is also reasonably well-written. -- I read the author response. Something that might make the paper more useful to a practical reader is a more concrete statement of what the results mean for Renyi-DP -- another related relaxation. Finally, a related paper is [1], which also does a Renyi-DP based analysis for smoothed sensitivity algorithms. It would be nice to see how the results in this paper compare with [1]. [1] Scalable Private Learning with PATE Nicolas Papernot, Shuang Song, Ilya Mironov, Ananth Raghunathan, Kunal Talwar, Ăšlfar Erlingsson, ICLR 2018.

Reviewer 2



I thank the authors for shedding some light on how these results compare with Karwa and Vadhan [15], and Feldman and Steinke [8]. The results on mean estimation however do seem narrow in scope. The paper will be stronger if the authors could discuss maybe other applications of the smooth sensitivity and CDP theory. --------------- The paper extends the smooth sensitivity framework of Nissim et al. [18] by identifying three new distributions from which additive noise scaled to smooth sensitivity provides concentrated differential privacy. These techniques are applied to the problem of mean estimation (for data drawn i.i.d. from a distribution) using a trimmed mean estimator. The authors present two results for this mean estimation problem. The first gives a stronger accuracy guarantee under symmetric subgaussian assumption, whereas the second gives a weaker accuracy guarantee under minimal distributional assumptions. The authors also present an experimental evaluation of these techniques on Gaussian data. The theoretical results appear correct. The identification of the distributions that provide CDP when scaled to smooth sensitivity could be useful. The main results in this paper are the CDP bounds for estimating the mean. In this case, Theorem 7 (on general distributions) looks interesting. However, it is unclear how significant is the improvement over the existing results of Feldman and Steinke [8]. Overall, at this point it feels like the current results are not significant enough for NeurIPS.

Reviewer 3



This paper investigate the problem of private mean estimation. Specifically, authors exploit the smooth sensitivity and concentrated differential privacy( CDP), and then provide CDP for three distribution with quasi-polynomial tails when scaled to smooth sensitivity.

[Author Response · NeurIPS 2019]

We thank all three reviewers for their comments, which we respond to below.

• Reviewers 2 and 3 ask about the comparison to the prior mean estimation work of Feldman and Steinke [8] and the
truncation/global sensitivity approach of Karwa and Vadhan [15]. We agree that a more detailed comparison to these
works is needed and we will add this to our paper. We briefly describe the advantages of our approach now.

In short, our algorithm is applicable to a wide variety of distributions, whereas the prior works lack this versatility.

The truncation/global sensitivity approach of Karwa and Vadhan works for light-tailed distributions like the Gaussian.
However, the accuracy rapidly degrades if the distribution has heavier tails (e.g., a Student's T distribution or a log-
Normal); the truncation interval must be very large to avoid biasing the output, but this results in high global sensitivity
and thus more variance due to added noise. Our trimming approach is much more resilient to heavy tailed distributions
without sacrificing anything in the ideal case where the distribution is light-tailed.

We also consider global sensitivity as a comparison point in our experimental evaluation. (This is the flat line in the
plots.) The relative performance depends on the truncation interval $[a, b]$. For the given parameters, smooth sensitivity
performs much better.

On the other hand, the median-of-means approach of Feldman and Steinke is more resilient to heavy tails. However,
their approach cannot adapt when the distribution is well-behaved. For example, if the data is Gaussian, then our
algorithm attains near-optimal accuracy per Theorem 6, whereas their algorithm cannot asymptotically match this level
of accuracy for any setting of its parameters.

We remark that our results for differential privacy are actually formally incomparable to those of Feldman and Steinke.
Their results are in the context of adaptive data analysis, rather than privacy. In particular, they do not provide a
comparable theorem statement and the algorithm is only implicit in their work. Thus the comparison we draw in this
discussion is based on our interpretation and analysis of their approach, rather than their stated results.

Generally, our approach can handle heavy or light tails with ease because the trimming automatically adapts to the
distribution (unlike truncation).

We will clarify these points in the paper by adding a theorem statement covering classes of distributions for which our
approach greatly outperforms both prior approaches – e.g., heavy-tailed symmetric distributions, such as the Student's
T.

• Reviewer 2 asks about other uses of the smooth sensitivity framework. We describe a few examples in the related work
section. (Since the NeurIPS submission deadline, two further papers have appeared that apply the smooth sensitivity
framework to median estimation and to estimating the degree distribution of a graph.) Implementing these is beyond
the scope of our current work.

We remark that the adoption of smooth sensitivity in applied work has been hampered by a few factors. One is
computational intractability for many problems, as noted by Reviewer 1. Another is the fact that the existing noise
distributions are unwieldy – either they have very heavy tails or a $\log(1/\delta)$ factor appears in their scale – which can be
fatal in practice. Our work addresses this situation by studying a fundamental building block (mean estimation) in the
design of differentially private algorithms for which computational intractability is not an issue, and by introducing
noise distributions with better properties.

• Reviewer 1 asks about Rényi differential privacy (RDP) and the comparison to the work of Papernot et al. using
Gaussian noise for smooth sensitivity under RDP. Our results are immediately applicable to RDP and we will add an
explicit discussion of this. In particular, CDP is equivalent to an infinite family of RDP guarantees which we will state
formally in the next version of the paper.

The RDP analysis of Gaussian noise in the work of Papernot et al. is similar to that for tCDP, which we already include
in our experimental evaluation as a comparison point. RDP potentially yields tighter constants in the analysis, relative
to tCDP. Since our experimental results suggest that Gaussian noise is not very promising, we did not investigate it
extensively.

[Meta-Review · NeurIPS 2019]

This paper re-visits the smooth sensitivity framework from the point of view of concentrated DP and proposes three new perturbation methods that work under this framework. The results are applied to the problem of estimating the mean of distributions with unbounded support from iid observations, and provide finite-sample bounds on the accuracy. This is a nice fundamental innovation in the theory of DP. When preparing a final version of the manuscript the authors are strongly encouraged to include a discussion about potential further applications of the new mechanisms beyond mean estimation.